# Deprotection of centromeric cohesin at meiosis II requires APC/C activity but not kinetochore tension

Valentina Mengoli[1,†,‡], Katarzyna Jonak[1,†] (iD), Oleksii Lyzak[1,†] (iD), Mahdi Lamb[2] (iD), Lisa M Lister[2], Chris Lodge[2], Julie Rojas[1] (iD), Ievgeniia Zagoriy[1,§] (iD), Mary Herbert[2,*] (iD) & Wolfgang Zachariae[1,**] (iD)

## Abstract

**Genome haploidization involves sequential loss of cohesin from chromosome arms and centromeres during two meiotic divisions. At centromeres, cohesin's Rec8 subunit is protected from separase cleavage at meiosis I and then deprotected to allow its cleavage at meiosis II. Protection of centromeric cohesin by shugoshin-PP2A seems evolutionarily conserved. However, deprotection has been proposed to rely on spindle forces separating the Rec8 protector from cohesin at metaphase II in mammalian oocytes and on APC/C-dependent destruction of the protector at anaphase II in yeast. Here, we have activated APC/C in the absence of sister kinetochore biorientation at meiosis II in yeast and mouse oocytes, and find that bipolar spindle forces are dispensable for sister centromere separation in both systems. Furthermore, we show that at least in yeast, protection of Rec8 by shugoshin and inhibition of separase by securin are both required for the stability of centromeric cohesin at metaphase II. Our data imply that related mechanisms preserve the integrity of dyad chromosomes during the short metaphase II of yeast and the prolonged metaphase II arrest of mammalian oocytes.**

**Keywords** biorientation; centromeric cohesion; meiosis II; protection; shugoshin

**Subject Category** Cell Cycle

**The EMBO Journal (2021) 40: e106812**

See also: **Y Gryaznova et al** (April 2021)

## Introduction

Accurate chromosome segregation depends on the ability of kinetochores to translate mechanical tension into biochemical signals. Tension arises between and within bioriented sister kinetochores, which are pulled toward opposite spindle poles by microtubules, while their centromeres are held together by cohesin complexes that entrap sister DNAs within a proteinaceous ring. In the absence of tension, kinetochores detach from microtubules and activate the spindle-assembly checkpoint (SAC), a signaling cascade that generates a diffusible inhibitor, called the mitotic checkpoint complex (MCC), of the ubiquitin-ligase APC/C$^{Cdc20}$ (Musacchio, 2015). MCC production ceases when all kinetochores are under tension, enabling APC/C$^{Cdc20}$ to promote entry into anaphase by targeting for degradation B-type cyclins and Pds1/securin, the inhibitor of the separase protease. Separase then triggers chromatid disjunction by cleaving the kleisin subunit of cohesin (Uhlmann et al, 2000).

Sister chromatid cohesion established during premeiotic S phase mediates the disjunction of maternal and paternal centromeres at meiosis I and the segregation of sister centromeres at meiosis II. This is possible because meiotic recombination creates bivalent chromosomes wherein cohesins connect not only sister centromeres but also, via chiasmata, maternal and paternal centromeres (Petronczki et al, 2003). In addition, meiosis I-specific activities, such as monopolin in budding yeast, cause sister kinetochores to function as a single, maternal or paternal kinetochore (Nasmyth, 2015). Pulling these kinetochores to opposite spindle poles now generates the tension required for silencing the SAC (Hauf & Watanabe, 2004). At anaphase I, separase cleaves cohesin on chromosome arms, which triggers the resolution of chiasmata and the segregation of dyad chromosomes (Buonomo et al, 2000; Kudo et al, 2006). Dyad chromosomes consist of two chromatids linked at their centromeres by cohesin that has escaped cleavage by separase at anaphase I and now serves to biorient sister centromeres on the meiosis II spindle. At anaphase II, a second wave of separase activity disjoins sister centromeres, leading to the formation of nuclei with a haploid genome.

The persistence of centromeric cohesin beyond anaphase I depends on its meiosis-specific kleisin subunit Rec8 (Klein et al, 1999; Watanabe & Nurse, 1999), which requires phosphorylation in order to be cleaved by separase. In yeast, the conserved kinases Hrr25/CK1δ and Cdc7-Dbf4 phosphorylate Rec8 at meiosis I (Ishiguro et al, 2010; Katis et al, 2010). Meanwhile, centromeric Rec8 is dephosphorylated and thereby protected from separase by the PP2A

1  Laboratory of Chromosome Biology, Max Planck Institute of Biochemistry, Martinsried, Germany
2  Biosciences Institute, Centre for Life, Times Square, Newcastle University, Newcastle upon Tyne, UK
   *Corresponding author. Tel: +44 191 213 8213; E-mail: mary.herbert@newcastle.ac.uk
   **Corresponding author. Tel: +49 89 8578 3105; E-mail: zachar@biochem.mpg.de
   †These authors contributed equally to this work
   ‡Present address: Institute for Research in Biomedicine, Università della Svizzera Italiana, Bellinzona, Switzerland
   §Present address: EMBL Heidelberg, Heidelberg, Germany

phosphatase containing the regulatory Rts1/B56 subunit, which is recruited to centromeres by a shugoshin protein (Kitajima *et al*, 2006; Riedel *et al*, 2006). Shugoshins have additional functions, including the formation of tension-generating microtubule-kinetochore attachments and, in mammalian mitosis, protection of centromeric cohesin from removal by a separase-independent mechanism, called the prophase pathway (Gutierrez-Caballero *et al*, 2012; Marston, 2015). Whereas budding yeast contains a single shugoshin (Sgo1), mammals encode two paralogues: Sgol1 protects centromeric cohesin from the prophase pathway at mitosis, and Sgol2 protects it from separase at anaphase of meiosis I (McGuinness *et al*, 2005; Llano *et al*, 2008).

While the role of shugoshin-PP2A in protecting centromeric Rec8 is evolutionarily conserved, it has remained unclear how and when centromeric Rec8 is "deprotected" to facilitate its phosphorylation and cleavage at anaphase II. Recent work in budding yeast supports a model, called deprotection-by-APC/C, whereby APC/C$^{Cdc20}$ activity removes PP2A from centromeres at anaphase II by mediating the degradation of Sgo1 and the Mps1 kinase, which is required for Sgo1's binding to centromeres. Centromeric Rec8 is then phosphorylated by Hrr25 (Arguello-Miranda *et al*, 2017; Jonak *et al*, 2017). This model coordinates deprotection with separase activation via APC/C$^{Cdc20}$ and therefore implies that centromeric Rec8 remains protected until entry into anaphase II. While the deprotection-by-APC/C model does not readily explain how deprotection of centromeric cohesin is confined to meiosis II, it seems appealing for mammalian oocytes, which arrest at metaphase II between ovulation and fertilization. The metaphase II arrest depends on inhibition of APC/C$^{Cdc20}$ by the cytostatic factor (CSF) Emi2, which is inactivated by a surge in cytoplasmic Ca$^{2+}$ levels in response to sperm entry (Madgwick *et al*, 2006; Shoji *et al*, 2006). However, current models for deprotection of Rec8 in mouse oocytes differ markedly in the mechanism and timing of Rec8 deprotection.

A longstanding model, herein called deprotection-by-tension, proposes that bipolar spindle forces spatially separate Sgol2-PP2A from centromeric cohesin, which exposes Rec8 to phosphorylation by the mammalian Rec8-kinase (Gomez *et al*, 2007; Lee *et al*, 2008). Thus, tension is assumed to first deprotect Rec8 as individual chromosomes biorient on the meiosis II spindle and then to silence the SAC when all chromosomes have achieved biorientation. In contrast to deprotection-by-APC/C, the deprotection-by-tension model offers a hypothesis for how deprotection is restricted to meiosis II. On the other hand, this model deprotects centromeric Rec8 at a time when protection might be deemed important, namely during the prolonged metaphase II arrest of the oocyte. Significantly, deprotection-by-tension has not been tested in meiotic cells.

Another oocyte study proposed that the conserved histone chaperone SET/TAF-1β inhibits PP2A at centromeres at meiosis II (Chambon *et al*, 2013). How SET affects PP2A's activity and how it does so specifically at meiosis II is unclear. Moreover, the SET orthologues of budding yeast are dispensable for meiotic chromosome segregation (Jonak *et al*, 2017). By contrast, spatial separation of Sgo1-PP2A and Rec8 at metaphase II has also been observed on chromosome spreads from yeast (Katis *et al*, 2004; Arguello-Miranda *et al*, 2017). This raises the question of whether kinetochore tension promotes the removal of centromeric cohesin solely by regulating the activity of APC/C$^{Cdc20}$ or by functioning both upstream and downstream of APC/C$^{Cdc20}$. In mitotic cells, Sgo1 is removed from kinetochores in a tension-dependent, but APC/C-independent manner as chromosomes biorient on the spindle at metaphase. Sgo1 released into the nucleoplasm is then degraded by APC/C-dependent proteolysis at anaphase (Eshleman & Morgan, 2014; Nerusheva *et al*, 2014). Likewise, the biochemical properties of Mps1 are known to change in response to tension (Maure *et al*, 2007; Aravamudhan *et al*, 2015). The question of whether tension plays a role beyond SAC silencing in centromeric cohesin removal could be addressed by activating APC/C$^{Cdc20}$ in the absence of sister kinetochore biorientation at meiosis II. However, such an experiment has neither been reported for mammalian nor for yeast meiosis.

Here, we have analyzed the fate of centromeric cohesion upon APC/C activation in the absence of sister kinetochore biorientation at meiosis II in yeast and mouse oocytes. In yeast, APC/C$^{Cdc20}$ activity induces removal of centromeric cohesin in a mutant unable to form functional spindles at meiosis II, which is inconsistent with a deprotection-by-tension mechanism. Furthermore, experimental activation of separase in cells arrested at metaphase II due to the lack of APC/C activity revealed that centromeric Rec8 is protected in the presence of bioriented sister kinetochores. Our data also argue against a deprotection-by-tension mechanism in metaphase II-arrested oocytes because APC/C$^{Cdc20}$ activated in the presence of nocodazole elicits sister centromere separation. We conclude that in oocytes, as in yeast, the sole essential role of kinetochore biorientation in cohesin cleavage is to activate APC/C$^{Cdc20}$. Thus, centromeric Rec8 might be protected during the metaphase II arrest of oocytes.

## Results

### A strategy to disrupt spindle function at meiosis II in yeast

To investigate the role of bipolar spindle forces in the deprotection of centromeric Rec8 in yeast, we sought to prevent sister kinetochores from biorienting at meiosis II. We used a genetic approach since microtubule-depolymerizing drugs have detrimental side effects at the doses required for destroying spindles in meiotic cells (Hochwagen *et al*, 2005). We hypothesized that spindle pole bodies (SPBs), the yeast centrosome equivalents, fail to assemble a functional spindle if they are too far apart. SPBs duplicate in S phase and separate upon formation of the meiosis I spindle at metaphase I. SPB reduplication at metaphase II then creates the poles of the two meiosis II spindles (Moens & Rapport, 1971) (Fig 1A, left). If SPBs cannot reduplicate at metaphase II, the now widely separated meiosis I-SPBs nucleate two half-spindles, which might fail to fuse into a bipolar structure capable of biorienting sister kinetochores (Fig 1A, right).

It has been reported that the *spo12Δ* mutant fails to reduplicate SPBs (Buonomo *et al*, 2003; Marston *et al*, 2003), which we confirmed by live-imaging of GFP-marked SPBs and RFP-tagged tubulin (Fig EV1A). As a result, these cells contain a single SPB per nucleus after anaphase I. While a normal-looking spindle initially appears in *spo12Δ* cells, an intact spindle is rarely observed at the time when wild-type cells contain a pair of meiosis II spindles. To distinguish meiosis I and II in *spo12Δ* cells, we imaged Rec8-GFP together with RFP tubulin (Fig 1B). Relative to cohesin cleavage at anaphase I, *spo12Δ* cells assemble and elongate a meiosis I spindle with normal kinetics. However, after the appearance of centromeric Rec8,

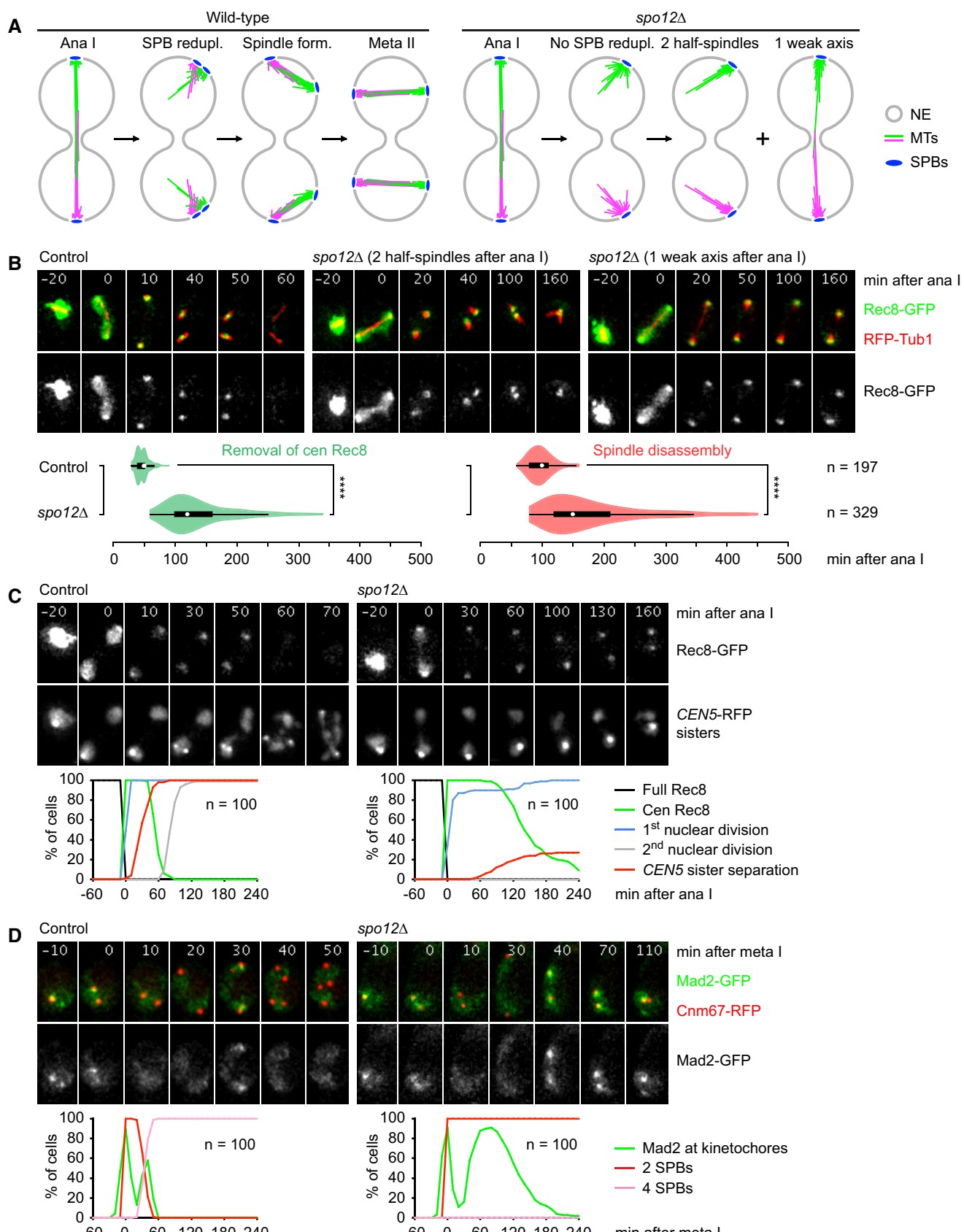

**Figure 1.**

**Figure 1.  *spo12Δ* cells fail to biorient sister kinetochores at meiosis II.**

A   Cartoon of microtubule structures at meiosis II in the wild-type and a mutant (*spo12Δ*) defective in SPB reduplication. NE, nuclear envelope; MTs, microtubules.
B   Imaging of Rec8-GFP and RFP-tubulin in control and *spo12Δ* cells. Top, time-lapse series. Bottom, times from cohesin cleavage at anaphase I to the removal of centromeric Rec8 and to spindle disassembly. ****$P < 0.0001$, Mann–Whitney (M–W) test.
C   Imaging of Rec8-GFP and *CEN5*-RFP sister dots in control and *spo12Δ* cells. Top, time-lapse series. Bottom, quantification of meiotic events in cells synchronized in silico to anaphase I (Rec8 cleavage) at *t* = 0.
D   Imaging of Mad2-GFP and SPBs (Cnm67-RFP) in control and *spo12Δ* cells. Top, time-lapse series. Bottom, quantification of the presence of Mad2-GFP foci in cells synchronized in silico to metaphase I (SPB separation) at *t* = 0.

Data information: (B) and (C) are representative of three independent experiments.
Source data are available online for this figure.

*spo12Δ* cells show either two apparently unconnected half-spindles (~ 45%) or a spindle axis with a weak and often barely detectable midzone (hereafter called a weak spindle axis, ~ 55%). Thus, the two SPBs of *spo12Δ* cells assemble a normal spindle axis at meiosis I but cannot do so at meiosis II. As wild-type cells enter metaphase II, centromeric Rec8 accumulates as a dot in the middle of each spindle (Fig 1B) and therefore localizes midway between the kinetochore clusters at the spindle poles (Fig EV1B). This configuration is consistent with recent work in mitotic cells, suggesting that sister centromeres are held together by cohesin accumulating at pericentromeric sites on either side of the centromere (Paldi *et al*, 2020). Upon biorientation, sister kinetochores are pulled to the spindle poles while sister DNAs are connected by pericentromeric cohesin that now localizes midway between the poles. In *spo12Δ* cells, centromeric Rec8 signals either colocalize with the kinetochore clusters at the spindle poles (~ 60%) or start to spread along the spindle axis but rarely concentrate between the poles (Figs 1B and EV1B). Thus, sister centromeres remain attached to a single spindle pole after anaphase I, which is consistent with a defective, discontinuous spindle axis at meiosis II in *spo12Δ* cells.

## *spo12Δ* cells fail to biorient sister centromeres at meiosis II

Bipolar spindle forces are capable of separating centromeric sister DNAs despite the presence of cohesin. To visualize this state of tension, we imaged Rec8-GFP together with RFP fused to *tet* repressor, which binds to *tet* operators at the centromeres of one copy of chromosome V (*CEN5*-RFP sister dots) (Toth *et al*, 2000). In the wild-type, *CEN5*-RFP sister dots split into two signals in the presence of centromeric Rec8 in 92% of cells at metaphase II (Fig 1C). In *spo12Δ* cells, by contrast, splitting of *CEN5*-RFP sister dots in the presence of centromeric Rec8 is delayed and does not exceed 23%, although metaphase II is prolonged (see below). We conclude that in the *spo12Δ* mutant, sister centromere biorientation is either strongly reduced (cells with a weak spindle axis) or virtually absent (cells with two half-spindles). Hence, we can now study the localization of PP2A^Rts1 and centromeric Rec8 in the absence of bipolar spindle forces. Live imaging of control cells showed that Rts1 localizes to the four kinetochore clusters at entry into metaphase II (Fig EV1C). However, Rts1 largely disappears from kinetochores before cells enter anaphase II, around the time sister kinetochores biorient. In *spo12Δ* cells, by contrast, high levels of Rts1 continue to localize at kinetochores for long periods of time, either because kinetochores lack tension or because activation of the APC/C is delayed. Importantly, centromeric Rec8 signals now colocalize with the Rts1 foci at the kinetochore clusters (Fig EV1B and C). Thus,

bipolar spindle forces are required for separating the bulk of PP2A^Rts1 from centromeric Rec8 at metaphase II.

## *spo12Δ* cells prolong SAC activity and delay anaphase entry at meiosis II

In yeast, lack of kinetochore tension activates the SAC at mitosis and meiosis I (Shonn *et al*, 2000; Stern & Murray, 2001), but it was unclear whether the SAC is also operational at meiosis II. To monitor SAC activity, we fused GFP to the SAC protein Mad2, which binds to kinetochores that lack tension-generating microtubule attachments (Musacchio, 2015). Since kinetochores cluster at the spindle poles, we imaged Mad2-GFP together with RFP-labeled SPBs. In the wild-type, Mad2-GFP foci transiently appear in the vicinity of SPBs prior to each division, persisting for $19 \pm 6$ min at meiosis I and for $13 \pm 5$ min at meiosis II (Fig 1D). Foci formation correlates with Mad2's function in restraining entry into anaphase: The deletion of *MAD2* reduces the normal duration of metaphase I ($22 \pm 10$ min) by ~ 15 min and that of metaphase II ($40 \pm 11$ min) by ~ 10 min. Thus, Mad2-GFP foci reveal a transient activation of the SAC at both divisions of an unperturbed meiosis. In *spo12Δ* cells, Mad2-GFP foci formation at meiosis I is normal ($19 \pm 6$ min, $P = 0.53$, Mann–Whitney [M–W] test; Fig 1D). At meiosis II, however, Mad2-GFP foci formation is extended to $88 \pm 30$ min ($P < 0.0001$, M–W test), suggesting that *spo12Δ* cells prolong SAC activity at meiosis II but not at meiosis I.

To analyze progression through meiosis II, we measured the timing of anaphase II events relative to the removal of Rec8-GFP from chromosome arms at anaphase I. In control cells, centromeric Rec8 disappears at $47 \pm 10$ min, and spindles disassemble at $98 \pm 19$ min after anaphase I. In the *spo12Δ* mutant, by contrast, centromeric Rec8 persists for $138 \pm 54$ min and microtubule structures disappear at $178 \pm 78$ min after anaphase I (Fig 1B). Cells containing two half-spindles or a weak spindle axis behave similarly in this regard. Furthermore, we found that control cells remove Rts1 from kinetochores at $33 \pm 12$ min after anaphase I, while *spo12Δ* cells do so at $121 \pm 58$ min (Fig EV1D). These data suggest that metaphase II is prolonged in *spo12Δ* cells. Next, we detected APC/C substrates in protein extracts from cultures released from a prophase-arrest by an estradiol-inducible *NDT80* gene (Appendix Fig S1). At meiosis I, *spo12Δ* cells initiate APC/C-dependent proteolysis with normal kinetics, as judged from the disappearance of Rec8 and the meiosis I-specific protein Spo13 (Sullivan & Morgan, 2007). At meiosis II, however, the substrates of APC/C^Cdc20 (e.g., Clb1 and Sgo1) and those of the meiosis-specific APC/C^Ama1 (e.g., polo kinase Cdc5) are markedly stabilized. An obvious hypothesis is that the

SAC inhibits APC/C$^{Cdc20}$. This would delay, in turn, proteolysis mediated by APC/C$^{Ama1}$ whose activation at meiosis II depends on the activity of APC/C$^{Cdc20}$ (Oelschlaegel *et al*, 2005). However, Ama1 accumulates to high levels during the prolonged metaphase II of *spo12Δ* cells (Appendix Fig S1), which might cause these cells to eventually exit from meiosis.

## Restoring sister kinetochore biorientation at meiosis II in *spo12Δ* cells

While our data suggest that SAC activity delays the onset of anaphase II in *spo12Δ* cells, we cannot discount the possibility that other effects of the *SPO12* deletion contribute to this delay. For instance, Spo12 is required for the release of the Cdc14 phosphatase from its inhibitor in the nucleolus at anaphase (Buonomo *et al*, 2003; Marston *et al*, 2003). If SAC activity is the sole mechanism that delays entry into anaphase II in *spo12Δ* cells, restoring sister kinetochore biorientation should eliminate the delay. To test this, we sought to prevent *spo12Δ* cells from elongating the meiosis I spindle, which should keep the two SPBs closer together at anaphase I and thereby allow assembly of a functional spindle at metaphase II (Fig 2A). Thus, we removed the monopolin subunit Mam1, which causes sister kinetochores to biorient at meiosis I. As a result, the persistence of centromeric cohesin prevents nuclear division and spindle elongation at meiosis I (Toth *et al*, 2000). Indeed, *spo12Δ mam1Δ* double mutants show a normal-looking spindle at meiosis II (Fig 2B). Most of these cells (96%) split *CEN5*-RFP sister dots in the presence of centromeric Rec8, which accumulates in a single dot in the middle of the spindle (Fig 2B and C). Consistent with proper sister kinetochore biorientation, Mad2 resides at kinetochores at meiosis II for only $16 \pm 7$ min (Fig 2D). Accordingly, *spo12Δ mam1Δ* cells remove centromeric Rec8 and disassemble the spindle much earlier than *spo12Δ* cells, at $56 \pm 12$ min and $99 \pm 18$ min after anaphase I, respectively (Fig 2B). The timing of these events in *spo12Δ mam1Δ* cells is comparable to that in *mam1Δ* single mutants, implying that the *SPO12* deletion has little effect on the duration of metaphase II. Our data suggest that keeping the two SPBs of *spo12Δ* cells in close proximity (within one nucleus) at anaphase I restores sister kinetochore biorientation at meiosis II. Indeed, *spo12Δ mam1Δ* cells produce diploid spores with high viability (91%). Taken together, our data validate the *spo12Δ* single and the *spo12Δ mam1Δ* double mutant as tools to manipulate sister kinetochore biorientation at meiosis II.

Does meiosis I in monopolin mutants represent an alternative system to study the role of kinetochore tension in deprotection? While sister kinetochore biorientation prevents the meiosis I division in monopolin mutants, formation of tension-generating microtubule-kinetochore attachments is compromised. At meiosis I, Mad2-GFP foci persist for $19 \pm 6$ min in the wild-type but for $37 \pm 9$ min in *mam1Δ* cells ($P < 0.0001$, M–W test; Figs 1D and 2D), and metaphase I is extended in a Mad2-dependent manner in the mutant (Appendix Fig S2). Monopolin mutants might contain bivalent chromosomes with a mono-oriented and a bioriented pair of sister centromeres, resulting in spindle forces that cannot be balanced (Nerusheva *et al*, 2014). These findings argue that the role of sister kinetochore biorientation in deprotection be investigated at meiosis II.

## Removal of centromeric cohesin in the absence of sister kinetochore biorientation

Our data suggest that the SAC delays the activation of APC/C$^{Cdc20}$, and thereby that of separase, at meiosis II in *spo12Δ* cells. Disabling the SAC should therefore reveal whether APC/C$^{Cdc20}$ is or is not able to induce the removal of centromeric Rec8 in the absence of sister kinetochore biorientation. In the former case, deletion of *MAD2* in *spo12Δ* cells should advance the degradation of APC/C substrates and the removal of centromeric cohesin. In the latter case, only degradation of APC/C substrates should be advanced. First, we confirmed that sister kinetochore biorientation is essentially absent in *spo12Δ mad2Δ* cells. These cells hardly ever split *CEN5*-RFP sister dots in the presence of centromeric Rec8 (0.6%, Fig 3A). Next, we analyzed protein extracts from synchronized *spo12Δ mad2Δ* cells, revealing efficient degradation of APC/C substrates at anaphase II (Appendix Fig S3). Accordingly, deletion of *MAD2* in *spo12Δ* cells advances the time of spindle disassembly at meiosis II by ~ 75 min; it now occurs at $106 \pm 34$ min after anaphase I, which is close to the time observed in the wild-type (Fig 3B). Thus, disabling the SAC restores robust APC/C activity at meiosis II in *spo12Δ* cells. Consistent with efficient degradation of Sgo1, also the removal of Rts1 foci is advanced in *spo12Δ mad2Δ* cells, occurring at $46 \pm 12$ min after anaphase I (Fig 3C and D). This suggests that APC/C activity rather than sister kinetochore biorientation dissociates PP2A$^{Rts1}$ from kinetochores at meiosis II. Importantly, *spo12Δ mad2Δ* cells remove centromeric Rec8 ~ 80 min earlier than *spo12Δ* cells, at $59 \pm 17$ min after anaphase I (Fig 3B). Centromeric Rec8 disappears without a preceding phase of spatial separation from the bulk of PP2A$^{Rts1}$ during metaphase II (Fig 3C).

Sister centromeres are expected to separate upon cleavage of centromeric Rec8. However, this is difficult to detect in the small nuclei of living cells when bipolar spindle forces are absent. Chromatin spreads from *spo12Δ mad2Δ* cells confirmed that *CEN5*-GFP sister dots rarely split in the presence of centromeric Rec8 (3.0%) but separate, albeit inefficiently (51%), once Rec8 has been cleaved (Fig EV2A). Notably, *CEN5*-GFP sister dots stay close to each other at anaphase II. On the other hand, the distance between GFP dots marking sister sequences 394 kb away from the centromere is, on average, four times larger (Fig EV2B). This implies that even after the removal of centromeric cohesin, sister centromeres remain loosely connected in *spo12Δ mad2Δ* cells. We suspect that these connections consist of intertwined (catenated) sister DNAs, which arise as a consequence of DNA replication. While topoisomerase II (topo II) resolves most of the catenanes on chromosome arms before entry into M phase, catenanes at centromeres are resolved only after cohesin has been removed in a process facilitated by bipolar spindle forces (Wang *et al*, 2010; Farcas *et al*, 2011; Charbin *et al*, 2014). Taken together, our data suggest that in yeast, the essential function of bipolar spindle forces with regard to the cleavage of centromeric Rec8 is to activate APC/C$^{Cdc20}$ through silencing the SAC. While spatial separation of the bulk of PP2A$^{Rts1}$ and Rec8 accompanies sister kinetochore biorientation, it is dispensable for the deprotection of Rec8. Nevertheless, by promoting the resolution of sister centromere catenanes, bipolar spindle forces might have an important role in sister centromere disjunction after cohesin cleavage.

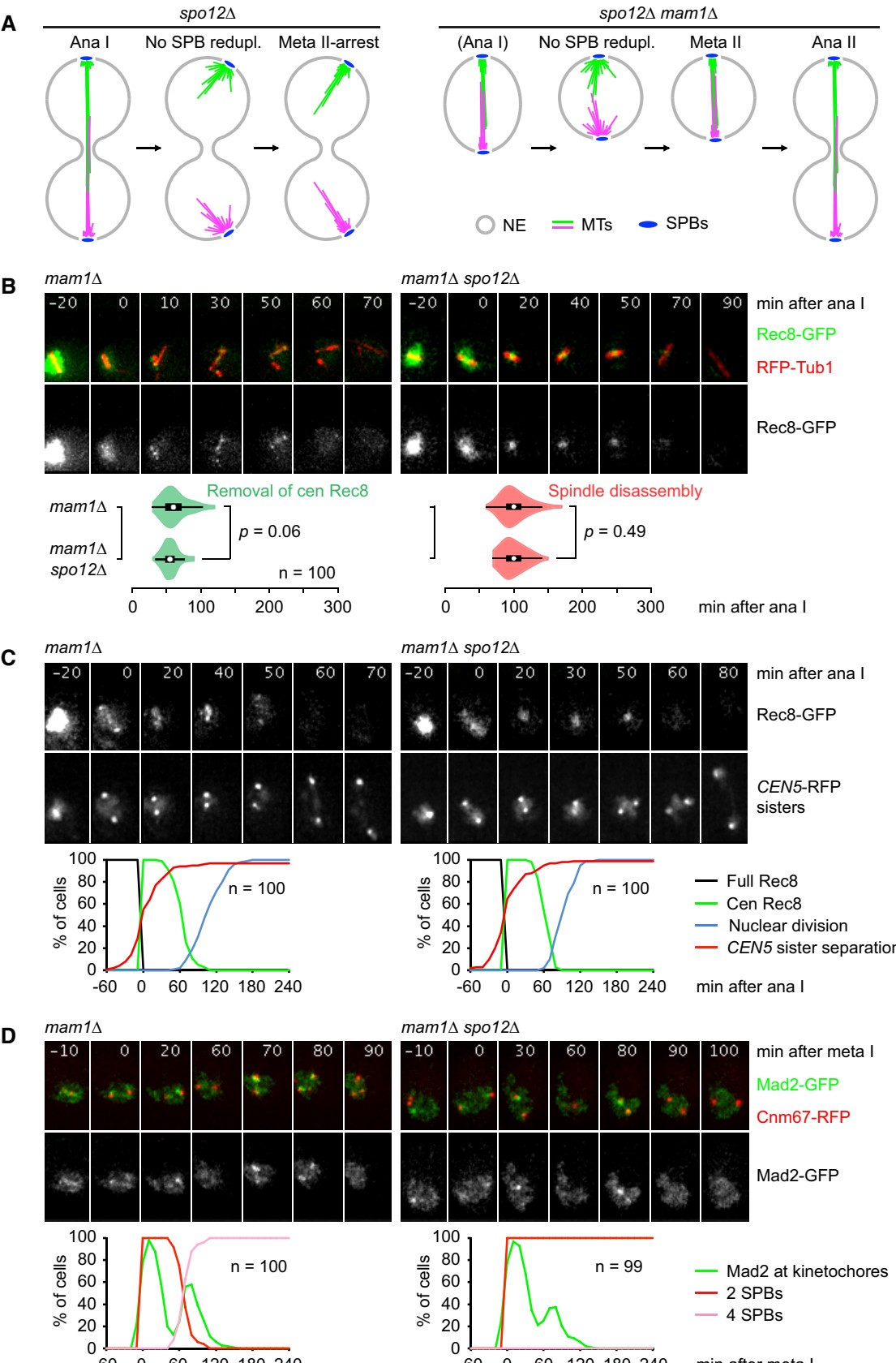

**Figure 2.**

◀

**Figure 2.  Deletion of monopolin restores sister kinetochore biorientation at meiosis II in *spo12Δ* cells.**

A  Cartoon showing how the *MAM1* deletion restores formation of a functional spindle at meiosis II in *spo12Δ* cells.
B  Imaging of Rec8-GFP and RFP-tubulin in *mam1Δ* and *mam1Δ spo12Δ* cells. Top, time-lapse series. Bottom, times from cohesin cleavage at anaphase I to the removal of centromeric Rec8 and to spindle disassembly. Data were compared using M–W tests.
C  Imaging of Rec8-GFP and CEN5-RFP sister dots in *mam1Δ* and *mam1Δ spo12Δ* cells. Top, time-lapse series. Bottom, quantification of meiotic events in cells synchronized in silico to anaphase I (Rec8 cleavage) at *t* = 0.
D  Imaging of Mad2-GFP and SPBs (Cnm67-RFP) in *mam1Δ* and *mam1Δ spo12Δ* cells. Top, time-lapse series. Bottom, quantification of the presence of Mad2-GFP foci in cells synchronized in silico to metaphase I (SPB separation) at *t* = 0.

Data information: Data are representative of three (B) or two (C) independent experiments.
Source data are available online for this figure.

## Tension regulates APC/C activity at meiosis II even in the absence of the MCC

The *SPO12* deletion causes a strong, Mad2-dependent delay in the onset of anaphase II. Surprisingly, it still delays entry into anaphase II, albeit for a shorter period of time, in the absence of Mad2: in the *mad2Δ spo12Δ* double mutant, the removal of Rts1 and Rec8 from centromeres and the disassembly of the spindle occur 20–30 min later than in the *mad2Δ* single mutant (Fig 3B and D). Accordingly, protein extracts from synchronized cultures show that APC/C-dependent proteolysis at meiosis II commences ~ 30 min later in *mad2Δ spo12Δ* than in *mad2Δ* cells (Appendix Fig S3). The *SPO12* deletion has a similar effect in cells that lack Mad1, Mad2, and Mad3 (Appendix Fig S4). Thus, the *SPO12* deletion delays activation of APC/C$^{Cdc20}$ in cells unable to assemble MCCs. While metaphase II is shorter in *mad2Δ* cells than in the wild-type, the mutant is capable of biorienting sister kinetochores, as revealed by frequent (69%) splitting of CEN5-RFP sister dots in the presence of centromeric Rec8. By contrast, such splitting is barely detectable (0.6%) in *mad2Δ spo12Δ* double mutants (Fig 3A). To test whether defective sister kinetochore biorientation delays entry into anaphase II even in the absence of Mad2, we sought to restore biorientation in *mad2Δ spo12Δ* cells by deleting *MAM1*. Indeed, the *mad2Δ spo12Δ mam1Δ* triple mutant splits CEN5-RFP sister dots in the presence of centromeric Rec8 as efficiently as the *mad2Δ* single mutant (67 vs. 69%, *P* = 0.75, Fisher's exact test; Fig 3A). Accordingly, the persistence of centromeric Rts1 and Rec8 and the time of spindle disassembly in the triple mutant are comparable to those observed in the *mad2Δ* single mutant (Fig 3B and D). Thus, deletion of *MAM1* restores sister kinetochore biorientation in *mad2Δ spo12Δ* cells, which restores, in turn, rapid entry into anaphase II. The *MAM1* deletion has a similar effect in *mad1,2,3Δ spo12Δ* cells (Appendix Fig S4). Another way to restore rapid entry into anaphase II in *mad2Δ spo12Δ* cells is to overexpress Cdc20 from an inducible promoter at meiosis II (Fig EV2C). Our data suggest that a mechanism, which does not require MCC formation, delays the activation of APC/C$^{Cdc20}$ when sister kinetochore biorientation is compromised at meiosis II. Such a mechanism has recently been described in worm embryos (Kim *et al*, 2017): Cdc20 is activated through dephosphorylation by the PP1 phosphatase, which occurs at kinetochores under tension (see discussion).

To investigate whether MCC-independent regulation of the APC/C exists at meiosis I, we analyzed *spo11Δ* mutants, which activate the SAC at meiosis I because they lack chiasmata connecting maternal and paternal centromeres (Shonn *et al*, 2000). Accordingly, the time from spindle formation to Rec8 cleavage (i.e., metaphase I) is significantly longer in *spo11Δ* cells than in the wild-type

(Appendix Fig S5). By contrast, the *SPO11* deletion has no detectable effect on the duration of metaphase I in cells lacking Mad2 (Appendix Fig S5). Thus, a tension-sensitive but MCC-independent regulation of APC/C activity is not apparent at meiosis I.

## A yeast strain for arrest/release at metaphase II

Our data suggest that kinetochore tension promotes deprotection of centromeric Rec8 by activating APC/C$^{Cdc20}$ via silencing the SAC, but has no essential role in deprotection downstream of APC/C$^{Cdc20}$. This notion predicts that centromeric Rec8 is protected when sister kinetochores are bioriented but the APC/C is inactive. To test this, we sought to activate separase in cells arrested at metaphase II with bioriented sister kinetochores. Thus, we constructed strains, called *cdc20$^{ts}$-mAR*, which initially arrest at metaphase I because the endogenous *CDC20* gene is controlled by a mitosis-specific promoter. Cells are released into anaphase I by expressing the temperature-sensitive *cdc20-3* allele from the copper-inducible *CUP1* promoter at 25°C. Shifting the culture to 37°C then inactivates Cdc20-3 and arrests cells at metaphase II. Being unable to generate APC/C activity, the Cdc20-3 proteins become stable and accumulate to high levels (Fig 4A, middle). Since strong accumulation of the APC/C activator Ama1 at later stages of meiosis might compromise the metaphase II arrest, *AMA1* was expressed from the early meiosis-specific *DMC1* promoter, resulting in *cdc20$^{ts}$-mAR ama1* cells.

At 37°C, *CDC20-mAR ama1* control cells, which express wild-type *CDC20*, enter anaphase II and exit from meiosis (Fig 4A and B, left). By contrast, shifting the *cdc20$^{ts}$-mAR ama1* strain to 37°C results in ~ 60% of metaphase II-arrested cells, which maintain high levels of APC/C substrates and a pair of short spindles for long periods of time (> 2 h; Fig 4A and B, middle). Remarkably, shifting the arrested cells back to 25°C causes prompt release into anaphase II, as judged from the degradation of APC/C$^{Cdc20}$ substrates, dephosphorylation of the Cdk1 substrate Ase1 (Khmelinskii *et al*, 2009), and a second nuclear division (Fig 4A and B, right). Live imaging confirmed that sister kinetochores are bioriented during the metaphase II arrest of *cdc20$^{ts}$-mAR ama1* cells (Fig 4C). While centromeric Rec8 localizes between kinetochore clusters, Rts1 localizes at kinetochore clusters for long periods of time. We conclude that APC/C$^{Cdc20}$ activity but not sister kinetochore biorientation removes PP2A$^{Rts1}$ from kinetochores at meiosis II.

## Protection of centromeric Rec8 at metaphase II in yeast

Next, we constructed *cdc20$^{ts}$-mAR ama1* strains in which separase can be activated in an APC/C-independent manner by auxin-

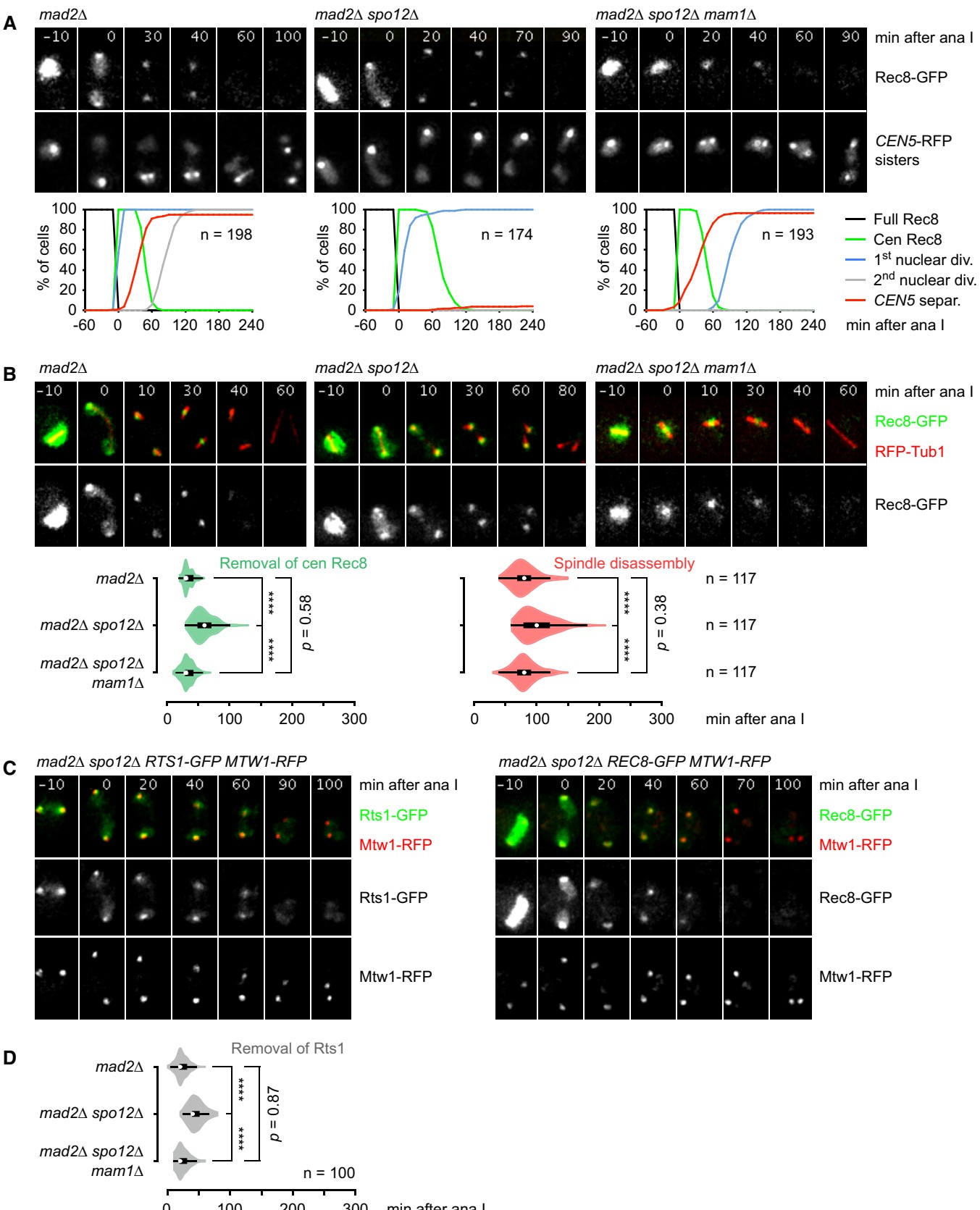

**Figure 3.**

**Figure 3.  APC/C activation in the presence and absence of sister kinetochore biorientation at meiosis II.**

A    Imaging of Rec8-GFP and *CEN5*-RFP sister dots in *mad2Δ*, *mad2Δ spo12Δ*, and *mad2Δ spo12Δ mam1Δ* cells. Top, time-lapse series. Bottom, quantification of meiotic events in cells synchronized in silico to anaphase I (Rec8 cleavage) at *t* = 0.

B    Imaging of Rec8-GFP and RFP-tubulin in *mad2Δ*, *mad2Δ spo12Δ*, and *mad2Δ spo12Δ mam1Δ* cells. Top, time-lapse series. Bottom, times from cohesin cleavage at anaphase I to the removal of centromeric Rec8 and to spindle disassembly. ****$P$ < 0.0001, M–W test.

C    Imaging of Rts1-GFP (left) or Rec8-GFP (right) and kinetochores (Mtw1-RFP) in the *mad2Δ spo12Δ* mutant.

D    Imaging of Rts1-GFP and Rec8-RFP was used to measure the times from cohesin cleavage at anaphase I to the removal of Rts1 from kinetochores in the indicated strains. ****$P$ < 0.0001, M–W test.

Data information: (A) and (D) are representative of two independent experiments.
Source data are available online for this figure.

inducible degradation (AID) of Pds1 (Nishimura *et al*, 2009). First, we tagged Pds1 with an AID domain. The resulting *PDS1-AID* cells serve as controls in which Pds1 degradation still requires the APC/C. Accordingly, these cells arrest at 37°C with high levels of Pds1-AID (Fig 5A). Consistent with sister kinetochores being bioriented, chromatin spreads show that centromeric Rec8 persists between spindle poles for long periods of time (apparent $t_{1/2}$ = 240 min; Fig 5B). By contrast, shifting control cells back to 25°C results in the degradation of Pds1-AID, the removal of centromeric Rec8 (initial $t_{1/2}$ = 22 min), and the formation of four equal-sized nuclei (Fig EV3A and B). To activate separase at the metaphase II arrest, we expressed the plant F-box protein OsTIR1 at anaphase I. Upon addition of auxin (IAA) at metaphase II, OsTIR1 targets Pds1-AID for rapid degradation (Fig 5A). This reduces the apparent half-life of centromeric Rec8 to 143 min (Fig 5B), suggesting that centromeric Rec8 is largely, though not perfectly, protected from separase.

Rec8 protection at metaphase II might depend on the Mps1 kinase and Sgo1 since co-expression of nondegradable Mps1 and Sgo1 hinders nuclear division at anaphase II (Arguello-Miranda *et al*, 2017). Thus, to deprotect Rec8, we first used *PDS1-AID* cells containing analogue-sensitive Mps1-as (Jones *et al*, 2005). Exposure of metaphase II-arrested cells to 1NM-PP1 reduces the apparent half-life of centromeric Rec8 to 147 min (Fig 5B), suggesting that separase is mostly, but not completely, inhibited in the presence of high levels of Pds1-AID. Accordingly, Rec8 rapidly disappears from chromatin (initial $t_{1/2}$ = 20 min), while cyclins remain stable, when inhibition of Mps1 is combined with depletion of Pds1-AID by OsTIR1 (Fig 5A and B). Next, we used OsTIR to degrade AID-tagged Sgo1 in metaphase II-arrested cells. While depletion of Sgo1-AID alone reduces the half-life of centromeric Rec8 to 60% of the control value (Fig 6A and B), depletion of Sgo1-AID together with Pds1-AID causes rapid removal of Rec8 (initial $t_{1/2}$ = 24 min; Fig 6C and D). Deprotection of Rec8 by inhibition of Mps1 or depletion of Sgo1 enables separase to elicit nuclear division. However, the resulting daughter nuclei are frequently of uneven size, probably because proper, anaphase-like chromosome movement requires the concomitant inactivation of Cdk1 (Oliveira *et al*, 2010). Taken together, our data suggest that two processes, protection of Rec8 and inhibition of separase, safeguard centromeric cohesin when APC/C is inactive and sister kinetochores are bioriented at metaphase II.

## Yeast cells cannot reinforce centromeric cohesion after anaphase I

Are protection of Rec8 and inhibition of separase the sole mechanisms safeguarding centromeric cohesion at metaphase II? A possibility worth considering is that meiotic cells replenish centromeric

cohesion at meiosis II. While sister chromatid cohesion is established during DNA replication, mitotic cohesin (containing Scc1/Rad21) can be reinforced at G2/M in response to DNA damage (Strom *et al*, 2007; Unal *et al*, 2007). To test whether centromeric cohesin can be reinforced at meiosis II, we used an estradiol-inducible promoter to express a non-cleavable version of Rec8 (Fig EV4). When expressed at entry into meiosis, the Rec8-24A mutant is incorporated into cohesin complexes that support normal sister chromatid cohesion. However, Rec8-24A is resistant to cleavage by separase due to the lack of phosphorylation sites for the Rec8-kinases Hrr25 and Cdc7-Dbf4. Thus, expression of Rec8-24A during S phase blocks nuclear division at meiosis I and -II in a dominant manner (Katis *et al*, 2010). By contrast, expression of Rec8-24A at anaphase I has no detectable effect on nuclear division at meiosis II (Fig EV4). It appears unlikely, therefore, that centromeric cohesin can be replenished after anaphase I in yeast.

## APC/C activity induces sister centromere separation in mouse oocytes lacking microtubules

In the deprotection-by-tension model originally proposed for metaphase II-arrested oocytes, bipolar spindle forces pull Sgol2 away from centromeric cohesin (Lee *et al*, 2008). However, it has not been tested whether spatial separation of Sgol2 and cohesin is actually required for the cleavage of centromeric Rec8 at anaphase II. Live imaging revealed that Sgol2 localizes between the kinetochores of dyad chromosomes in metaphase II-arrested oocytes (Fig EV5A). On chromosome spreads, Sgol2 forms a crescent at the centromeric end of each chromatid of a dyad chromosome (Fig EV5B). The two crescents extend to and overlap at the pericentromeric region, creating a prominent signal between the kinetochores. In addition, we noticed that Sgol2 persists around kinetochores as sister centromeres separate at the onset of anaphase II. However, detection of bulk Sgol2 might not reveal the behavior of the Rec8 protector, given that Sgol2 has multiple functions at centromeres (Rattani *et al*, 2013). We therefore sought a functional assay for the role of tension in sister centromere separation and investigated the consequences of activating APC/C$^{Cdc20}$ in the absence of bipolar spindle forces at meiosis II.

We harvested oocytes arrested at the germinal vesicle (GV) stage from mouse ovaries and allowed them to progress to the metaphase II arrest *in vitro*. In this arrest, sister centromeres are bioriented on the meiosis II spindle, and the SAC is inactive, while APC/C$^{Cdc20}$ is inhibited by the CSF Emi2 (Tsurumi *et al*, 2004). Emi2 can be inactivated by treatment with SrCl$_2$, which mimics the oscillations in cytosolic free Ca$^{2+}$ induced by sperm entry (Kline & Kline, 1992).

Accordingly, securin-mNeonGreen (securin-NG) expressed from microinjected mRNA accumulates in the arrested oocytes, but is degraded in the presence of SrCl₂ (Fig EV6A). Chromosome spreads prepared from arrested oocytes show 20 dyad chromosomes, which are converted to single chromatids within 60 min of exposure to SrCl₂ (Fig 7A). To facilitate scoring of sister centromere separation,

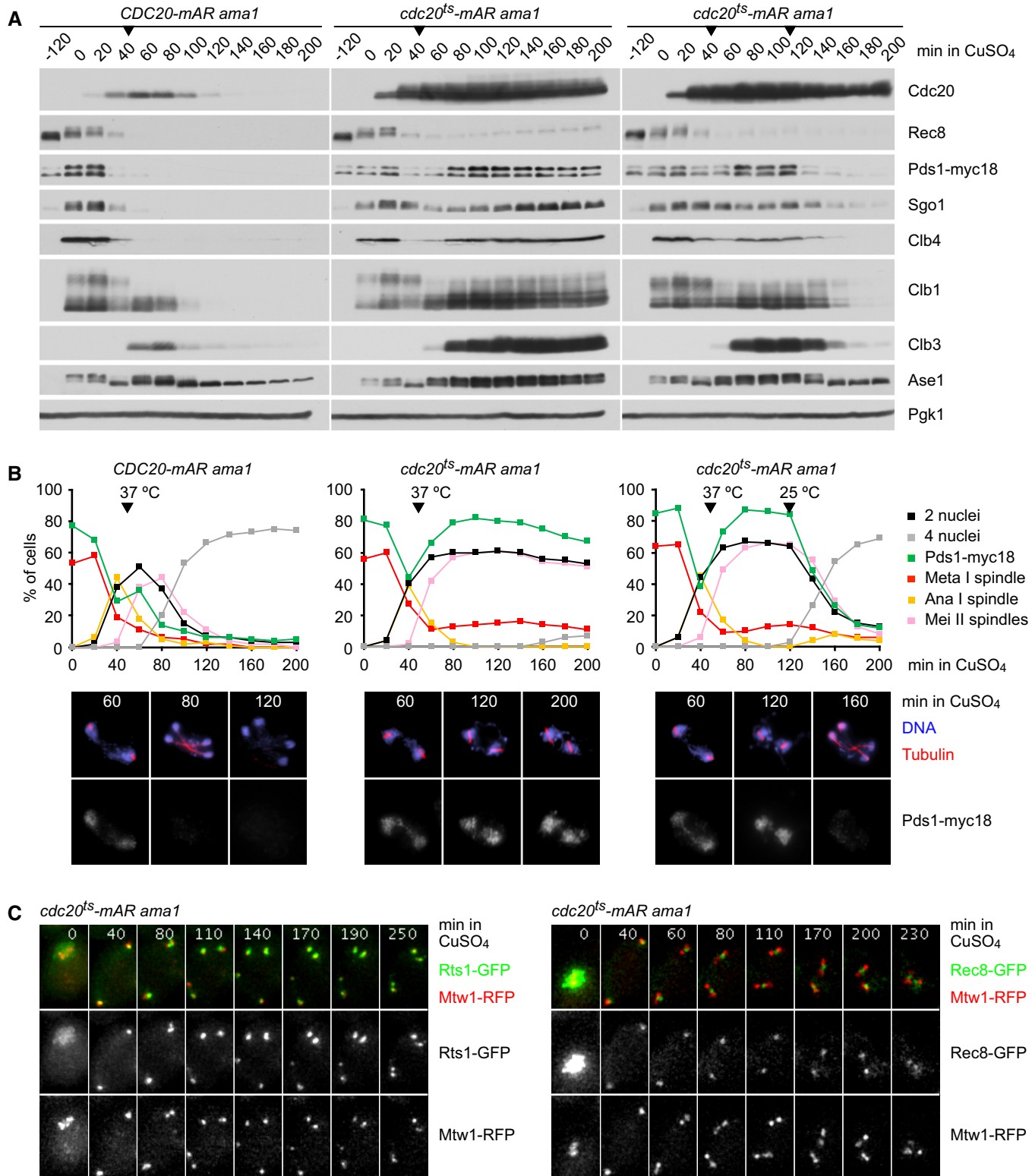

Figure 4.

**Figure 4. A yeast strain for arrest/release at metaphase II.**

A, B   Cells expressing endogenous *CDC20* from the mitosis-specific *CLB2* promoter and *AMA1* from the early meiosis-specific *DMC1* promoter were arrested at metaphase
  I at 25°C. At *t* = 0, cells were released into anaphase I by expressing either wild-type *CDC20* (*CDC20-mAR ama1*) or temperature-sensitive *cdc20-3* (*cdc20^ts^-mAR
  ama1*) from the *CUP1* promoter (+ CuSO₄). At *t* = 50 min, cultures were shifted to 37°C to arrest *cdc20^ts^-mAR ama1* strains at metaphase II. At *t* = 120 min, the
  culture on the right was shifted back to 25°C to release cells into anaphase II. (A) Immunoblot analysis of protein levels. (B) Top, quantification of cellular features
  by immunofluorescence microscopy of fixed cells (*n* = 100 per time point). Bottom, staining of DNA, spindles, and Pds1-myc18 in meiosis II cells.
C      Time-lapse series from the imaging of Rts1-GFP (left) or Rec8-GFP (right) and kinetochores (Mtw1-RFP) in *cdc20^ts^-mAR ama1* cells (*n* = 80) released into anaphase I
  at *t* = 0 and shifted from 25 to 37°C at *t* = 40 min.

Data information: Data are representative of three (B) or two (C) independent experiments.
Source data are available online for this figure.

we detected kinetochores together with topo II, which decorates the chromatid axis and accumulates around centromeres, as has been shown in spermatocytes (Gomez *et al*, 2007).

Next, we eliminated sister kinetochore biorientation by depolymerizing microtubules with nocodazole, which causes inhibition of APC/C^Cdc20 by the SAC. Under these conditions, Emi2 degradation in response to SrCl₂ fails to elicit APC/C^Cdc20 activity (Madgwick *et al*, 2006). Consequently, securin-NG accumulates (Fig EV6B), and chromosome spreads show 20 dyad chromosomes per oocyte (Fig 7B). By disabling the SAC, it should now be possible to activate APC/C^Cdc20 in SrCl₂-treated oocytes in the absence of kinetochore tension. To achieve this, we microinjected metaphase II-arrested oocytes with mRNA encoding either wild-type Cdc20 or Cdc20-R132A, which cannot bind Mad2 (Izawa & Pines, 2012). After allowing time for protein synthesis, oocytes were exposed to nocodazole and subsequently to SrCl₂. Under these conditions, wild-type Cdc20 is unable to induce degradation of securin-NG (Fig EV6C), and the incidence of separated centromeres remains low (2 ± 3%, Fig 7C). By contrast, oocytes expressing Cdc20-R132A readily degrade securin-NG (Fig EV6C) and display, on average, 45 ± 19% of separated centromeres (Fig 7C). Our data suggest that APC/C^Cdc20 activity can elicit sister centromere separation in the absence of microtubules, which is inconsistent with bipolar spindle forces playing an essential role in the deprotection of centromeric Rec8.

Nevertheless, nocodazole reduces the extent of sister centromere separation when we override the SAC using Cdc20-R132A. It appears that bipolar spindle forces promote sister centromere disjunction not only by silencing the SAC but also at a step downstream of APC/C^Cdc20. One possibility is that bipolar spindle forces, while not being essential, play an auxiliary role in deprotection of centromeric Rec8. Thus, we used the drug reversine to inhibit the Mps1 kinase whose activity is required for SAC signaling and the protection of centromeric Rec8 at anaphase I (Santaguida *et al*, 2010; El Yakoubi *et al*, 2017). In the presence of reversine, oocytes treated with nocodazole and SrCl₂ show degradation of securin-NG (Fig EV6B) and a marked increase in separated centromeres (46 ± 16%, Fig 7B), which is comparable to the value obtained upon expression of Cdc20-R132A (45 ± 19%, *P* = 0.88, M–W test). While these data confirm that APC/C is capable of inducing sister centromere separation in the absence of tension, inhibition of Mps1 does not further increase the efficiency of separation.

Another possibility is that centromeres remain connected even after Rec8 has been cleaved. At centromeres, sister DNA catenanes persist until after cohesin cleavage (Wang *et al*, 2010; Gomez *et al*, 2014). Importantly, bipolar spindle forces put catenanes under tension and thereby facilitate sister centromere decatenation by topo

II (Farcas *et al*, 2011; Charbin *et al*, 2014). Indeed, APC/C activity in the presence but not the absence of nocodazole gives rise to "loose dyads", pairs of chromatids with adjacent but seemingly disconnected centromeres (Fig 7C, bottom). To test whether a linkage other than cohesin hinders sister centromere disjunction upon APC/C activation in the presence of nocodazole, we used Trim-Away to destroy the Rec8 protein. Trim-Away utilizes the ubiquitin-ligase Trim21, which mediates the degradation of cytosolic antibodies together with the bound antigens (Clift *et al*, 2017). Thus, we injected GV-stage oocytes with mRNA encoding active Trim21 or a mutant deficient in antibody binding and allowed them to progress to metaphase II. As shown previously (Clift *et al*, 2017), injection of antibodies to Rec8 triggers efficient sister centromere separation in oocytes expressing active but not mutant Trim21 (98 ± 3 vs. 1 ± 2%, Fig 8A). By contrast, Trim-Away of Rec8 in oocytes treated with nocodazole results in only 48 ± 20% of separated centromeres (*P* < 0.0001, M–W test; Fig 8B). Similar to APC/C activity, Trim-Away of Rec8 generates "loose dyads" in the presence but not the absence of nocodazole.

Does Trim-Away of Rec8 destroy cohesin in the presence of nocodazole as efficiently as in its absence? To test this, we performed Trim-Away of Rec8 in the presence of nocodazole and then moved oocytes to nocodazole-free medium containing the proteasome inhibitor MG-132. This terminates Trim21-dependent proteolysis and allows reformation of the spindle, which then separates those dyads that have lost cohesin during the treatment with nocodazole. Indeed, the incidence of separated centromeres increases to 90 ± 8% (*P* < 0.0001, M–W test; Fig 8C). Our Trim-Away experiments suggest that even after complete destruction of cohesin, sister centromeres do not disjoin efficiently in the presence of nocodazole.

If sister centromere disjunction is hindered by incomplete decatenation of sister DNAs, it might be further reduced upon inhibition of topo II. In the presence of a functional spindle, the topo II inhibitor ICRF-193 has only a small effect on sister centromere separation but perturbs coordinated chromatid segregation to the spindle poles (Fig 9A). This is consistent with the notion that DNA catenanes cannot withstand bipolar spindle forces (Oliveira *et al*, 2010). By contrast, upon APC/C activation in the absence of microtubules (nocodazole + reversine + SrCl₂), ICRF-193 reduces sister centromere separation from 46 ± 12 to 22 ± 7% (Fig 9B). Taken together, our data show that in nocodazole-treated oocytes, the level of sister centromere separation elicited by activation of APC/C^Cdc20 approaches that achieved through Trim-Away of Rec8 (45 ± 19 vs. 48 ± 20%, *P* = 0.53, M–W test). We therefore conclude that in oocytes, bipolar spindle forces have no essential role beyond SAC silencing in the removal of centromeric cohesin at meiosis II.

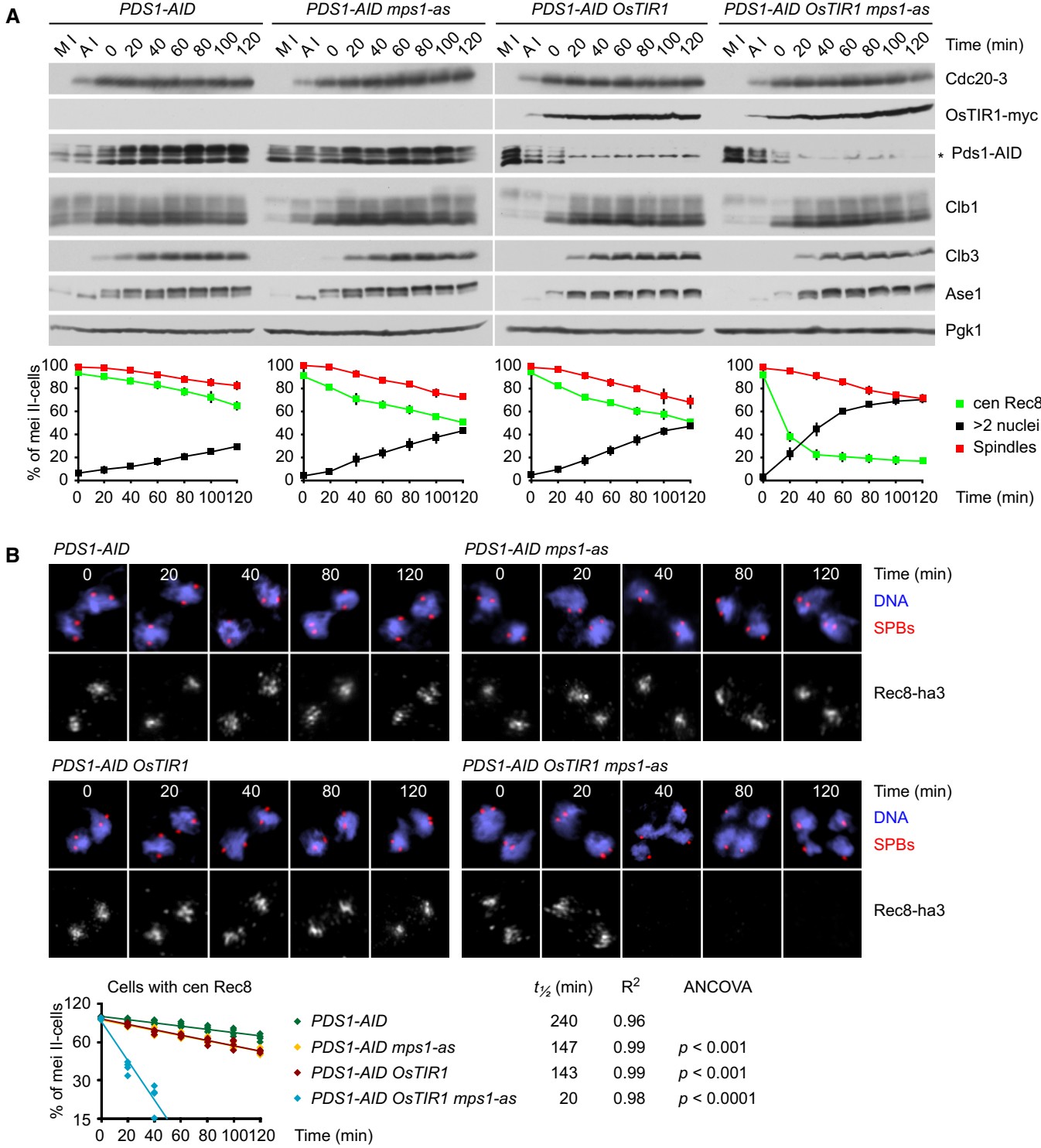

**Figure 5. Protection of centromeric Rec8 at metaphase II in yeast requires Mps1 activity.**

A, B cdc20ts-mAR ama1 PDS1-AID strains were released from metaphase I (M I) into anaphase I (A I) at 25°C and then arrested at metaphase II by a shift to 37°C. At t = 0, cultures were treated with 1NM-PP1 to inhibit Mps1 activity in mps1-as strains and with IAA to deplete Pds1-AID in strains expressing OsTIR1. (A) Top, immunoblot analysis of protein levels. The asterisk marks a nonspecific band. Bottom, percentages of meiosis II cells (four SPBs) with spindles, nuclear division (> 2 nuclei), and centromeric Rec8 (from chromatin spreads). Data are mean values ± s.d. of four independent experiments. (B) Top, chromatin spreads from meiosis II (four SPBs) stained for DNA, γ-tubulin/SPBs, and Rec8-ha3. Bottom, semi-log plot of the percentages of meiosis II cells with centromeric Rec8 from 4 independent experiments. Half-lives were calculated from exponential regression of the mean values. Slopes were compared with ANCOVA.

Source data are available online for this figure.

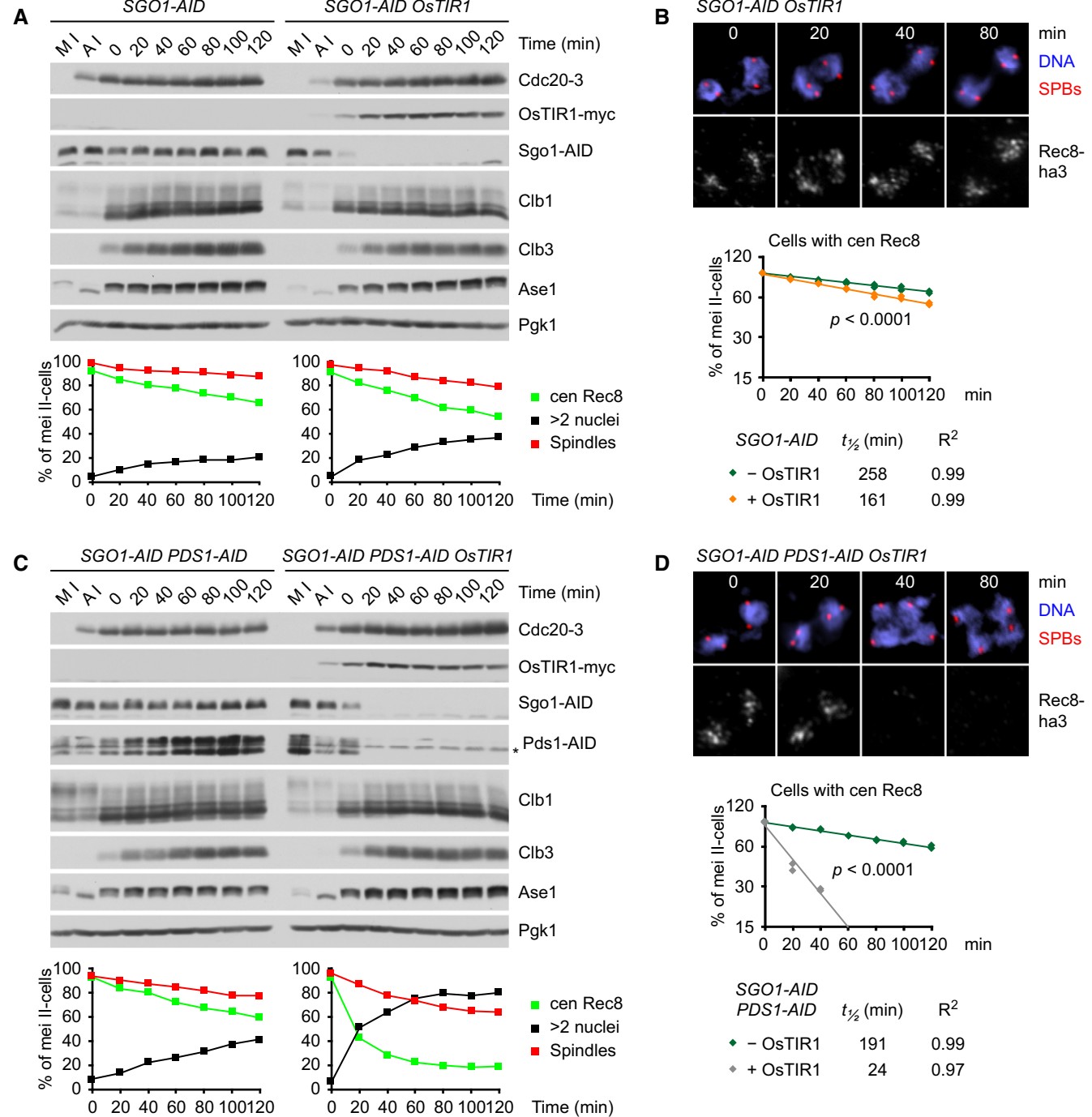

**Figure 6. Protection of centromeric Rec8 at metaphase II in yeast depends on Sgo1.**

A, B *cdc20^ts^-mAR ama1 SGO1-AID* strains were arrested at metaphase II and treated with IAA at *t* = 0 to deplete Pds1-AID in cells expressing OsTIR1. (A) Top, protein blots. Bottom, percentages (mean of two independent experiments) of meiosis II cells (four SPBs) with spindles, nuclear division (> 2 nuclei) and centromeric Rec8 (from chromatin spreads). (B) Top, chromatin spreads from meiosis II (four SPBs) stained for DNA, γ-tubulin/SPBs, and Rec8-ha3. Bottom, semi-log plot of the percentages of meiosis II cells with centromeric Rec8 from 2 independent experiments. Half-lives were calculated from exponential regression of the mean values. Slopes were compared with ANCOVA.

C, D *cdc20^ts^-mAR ama1 SGO1-AID PDS1-AID* strains were arrested at metaphase II as above and treated with IAA at *t* = 0 to deplete Sgo1-AID and Pds1-AID in cells expressing OsTIR1. (C) Protein blots and quantification of spindles, nuclear division, and centromeric Rec8 in meiosis II cells as in (A). (D) Chromatin spreads and half-lives of centromeric Rec8 as in (B).

Source data are available online for this figure.

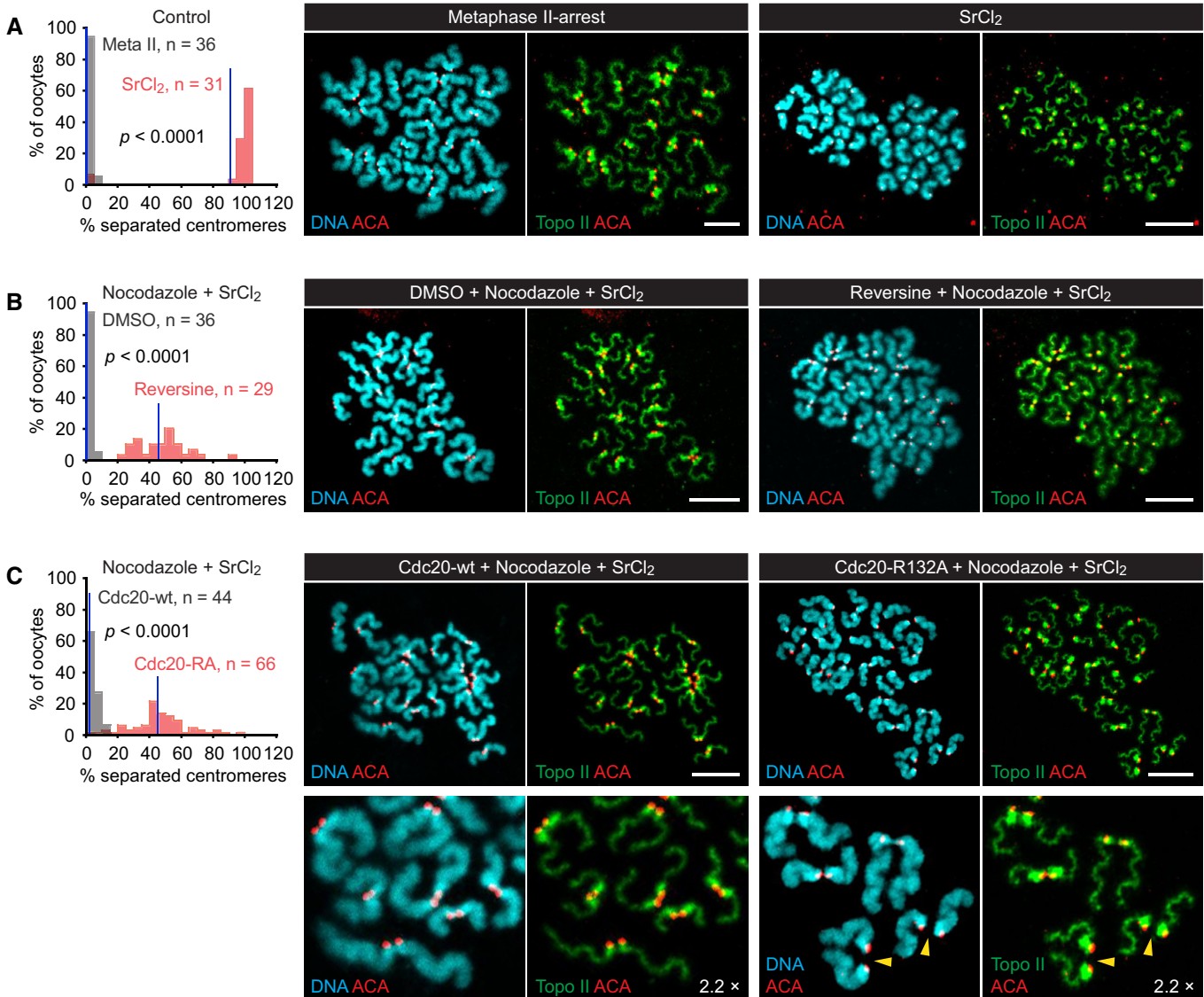

**Figure 7. Activation of APC/C in metaphase II-arrested oocytes containing or lacking microtubules.**

A–C  Metaphase II-arrested oocytes were treated (0.5 h) with DMSO (control) or nocodazole. When indicated, the SAC was disabled using reversine or expression of SAC-resistant Cdc20-R132A, and CSF was inactivated by addition of SrCl$_2$. Chromosome spreads were prepared 1 h after exposure to SrCl$_2$ and stained for DNA, kinetochores (ACA), and topo II. Histograms show percentages of separated centromeres per oocyte. Blue lines: mean values. Data were compared using M–W tests. Images show representative spreads. Scale bars: 10 μm. (A) Control oocytes treated with water or SrCl$_2$. (B) Oocytes treated with nocodazole or nocodazole plus reversine were exposed to SrCl$_2$. (C) Oocytes were allowed to express either Cdc20-wt or Cdc20-R132A from injected mRNA for 3 h and then exposed to nocodazole and SrCl$_2$. Bottom, cut-outs magnified 2.2×. Arrowheads mark loose dyads.

Source data are available online for this figure.

Nevertheless, spindle forces promote sister centromere disjunction after cohesin removal by facilitating the topo II-dependent decatenation of centromeric sister DNAs.

## Discussion

At centromeres, cohesin's Rec8 subunit is protected from cleavage by separase during anaphase I and then deprotected to allow its cleavage at anaphase II. While protection of Rec8 by Sgo-PP2A seems evolutionarily conserved, different models have been proposed for reversing protection. In the deprotection-by-APC/C model from yeast, APC/C$^{Cdc20}$ inactivates Sgo-PP2A concomitantly with the activation of separase. On the other hand, in the deprotection-by-tension mechanism proposed for mammalian meiosis, bipolar spindle forces deprotect centromeric Rec8 at metaphase II by pulling Sgol2-PP2A away from cohesin. Interestingly, each model has the potential to complement a weakness in the other. The yeast model is attractive for mammalian oocytes since it would protect Rec8 during the prolonged arrest at metaphase II. Conversely, the

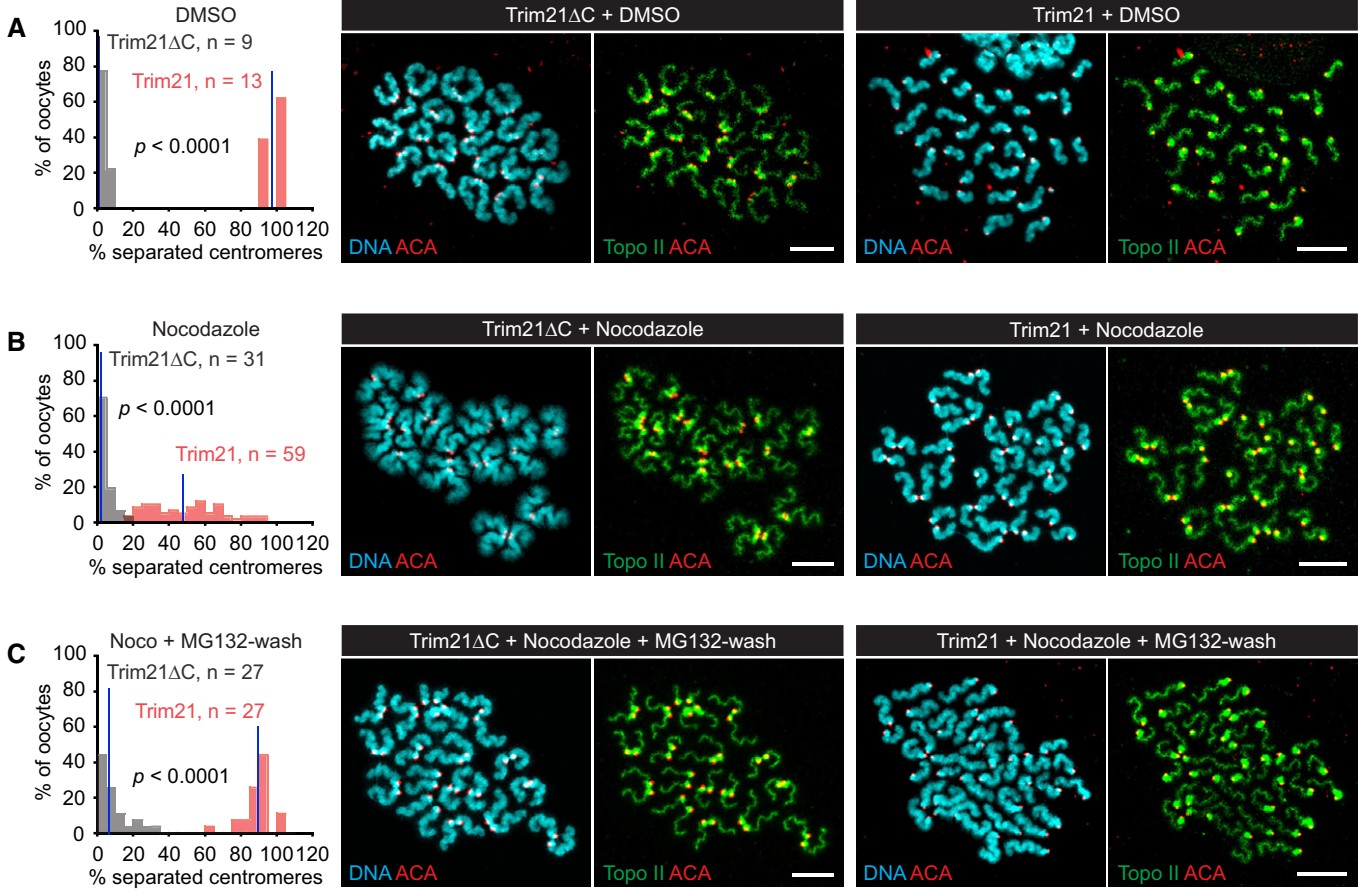

**Figure 8. Trim-Away of Rec8 in metaphase II-arrested oocytes containing or lacking microtubules.**

A–C GV-stage oocytes were injected with mRNA encoding active Trim21 or inactive Trim21ΔC, allowed to progress to metaphase II, and injected with α-Rec8 antibodies in the presence of DMSO (control) or nocodazole. Chromosome spreads prepared 1 h after antibody injection were stained for DNA, kinetochores (ACA), and topo II. Histograms show percentages of separated centromeres per oocyte. Blue lines: mean values. Data were compared using M–W tests. Images show representative spreads. Scale bars: 10 μm. (A) Control oocytes injected with antibodies in the presence of DMSO. (B) Oocytes exposed to nocodazole 0.5 h before antibody injection. (C) Oocytes injected with antibodies in the presence of nocodazole were moved to nocodazole-free medium containing MG-132 for 0.5 h before chromosome spreading.

Source data are available online for this figure.

deprotection-by-tension model might be relevant to yeast as it accommodates the separation of Sgo1-PP2A from cohesin observed at metaphase II and offers a hypothesis for how deprotection is confined to meiosis II. We have therefore tested in yeast and mouse oocytes a key prediction of the deprotection-by-tension model, namely that tension has two separate functions, SAC silencing and deprotection, in the removal of centromeric cohesin.

**The role of tension in chromatid segregation at meiosis II**

To prevent sister kinetochore biorientation at meiosis II in yeast, we used the *SPO12* deletion to block SPB/centrosome reduplication and thereby formation of a functional spindle at metaphase II. As a result, the SAC inhibits APC/C$^{Cdc20}$, which delays entry into anaphase II. Inactivation of the SAC restores not only rapid degradation of Cdc20 substrates but also the removal of centromeric cohesin. Centromeric Rec8 is cleaved without being separated from the bulk of PP2A$^{Rts1}$ during metaphase II. Thus, tension is

required for activation of APC/C$^{Cdc20}$ via silencing the SAC, but has no essential role in the cleavage of centromeric Rec8 downstream of APC/C$^{Cdc20}$. On the other hand, centromeric Rec8 is largely, though not completely, protected from separase when Pds1/securin is removed from cells arrested at metaphase II due to the lack of APC/C activity. This protection depends on Sgo1 and the kinase activity of Mps1, which is required for Sgo1's localization to kinetochores (Arguello-Miranda *et al*, 2017). Remarkably, the bulk of Sgo1-PP2A colocalizes with kinetochores that cluster at the spindle poles, while centromeric Rec8 accumulates in the middle of the spindle. These data suggest that in yeast, protection of centromeric Rec8 can coexist with sister kinetochore biorientation for long periods of time. Furthermore, our data imply that the regulation of Sgo1-PP2A at meiosis II differs from that at mitosis. In mitotic cells, Sgo1-PP2A disappears from kinetochores/centromeres when chromosomes biorient as cells arrest at metaphase due to the depletion of Cdc20 (Eshleman & Morgan, 2014; Nerusheva *et al*, 2014). At metaphase II, by contrast, Sgo1-

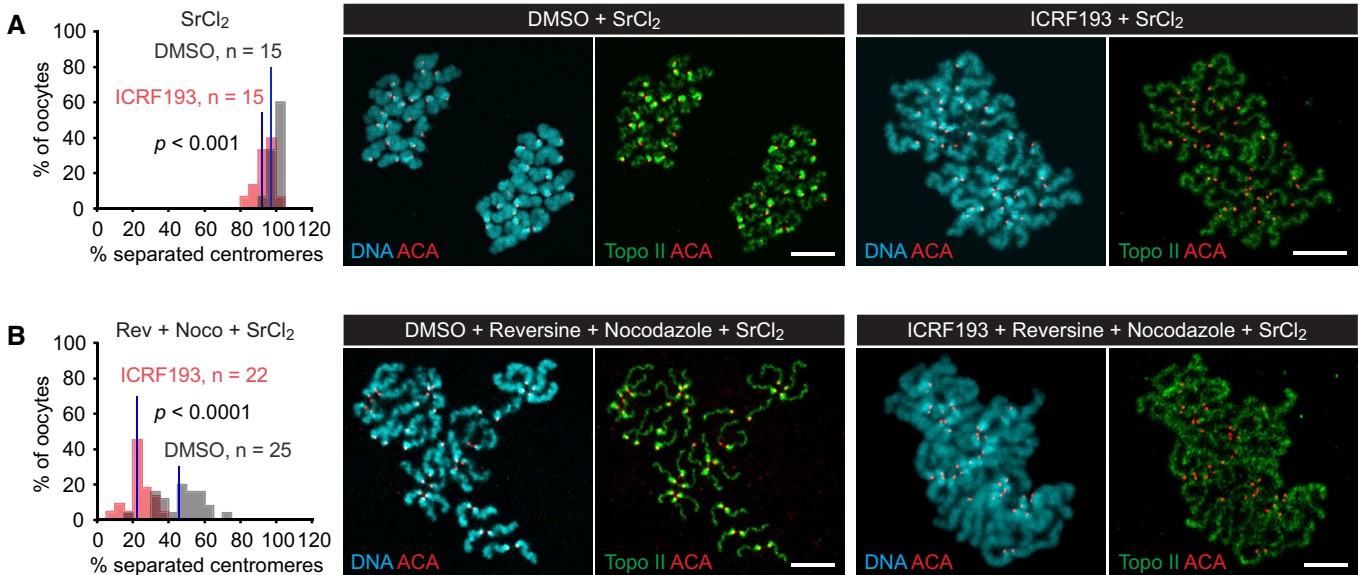

**Figure 9. The effect of topo II inhibition on sister centromere separation in oocytes containing a spindle or lacking microtubules.**

A, B  Metaphase II-oocytes were activated in medium containing $SrCl_2$ (A) or $SrCl_2$, nocodazole, and reversine (B) in the presence of DMSO or ICRF-193. Chromosome spreads prepared 1 h after activation were stained for DNA, kinetochores (ACA), and topo II. Histograms show percentages of separated centromeres per oocyte. Blue lines: mean values. Data were compared using M–W tests. Images show representative spreads. Scale bars: 10 μm.

Source data are available online for this figure.

PP2A persists at bioriented sister centromeres to ensure the protection of centromeric Rec8.

In oocytes, we used nocodazole to disrupt sister kinetochore biorientation during the metaphase II arrest and then released APC/$C^{Cdc20}$ from inhibition by the SAC and the CSF Emi2. Under these conditions, the deprotection-by-tension model predicts that separase is activated by APC/C-dependent proteolysis of Pds1/securin but fails to cleave centromeric Rec8 due to the lack of tension. We find, however, that APC/$C^{Cdc20}$ activity elicits sister centromere separation in the presence of nocodazole, suggesting that tension downstream of APC/$C^{Cdc20}$ is not essential for the cleavage of centromeric Rec8. Loss of sister centromere cohesion upon entry into anaphase II in the absence of bipolar spindle forces has also been observed in frog oocytes and insect spermatocytes (Paliulis & Nicklas, 2004; Shao *et al*, 2013). Independence from tension might be a conserved feature of the deprotection mechanism(s).

While dispensable for the cleavage of centromeric Rec8, tension downstream of APC/$C^{Cdc20}$ might nevertheless play a crucial role in sister centromere disjunction at anaphase II. In yeast and oocytes, removal of centromeric cohesin in the absence of tension gives rise to chromatids that behave as if loosely connected at their centromeres. During DNA replication, sister chromatids become connected not only by cohesins but also by the intertwining (catenation) of sister DNAs. While topo II resolves catenanes on chromosome arms before entry into M phase, it can complete the decatenation of sister centromeres only after cohesin has been cleaved (Wang *et al*, 2010; Farcas *et al*, 2011; Gomez *et al*, 2014). In principle, the strand-transfer reaction catalyzed by topo II can resolve as well as create catenanes (Nitiss, 2009). However, after cohesin has been cleaved, spindle forces separate sister centromere DNAs and put catenanes under tension, which drives the topo II reaction toward decatenation (Holm *et al*, 1989). Furthermore, topo II preferentially binds to the sharply bent DNA that tension creates at the catenation point (Vologodskii *et al*, 2001; Dong & Berger, 2007). Thus, decatenation of sister centromeres becomes inefficient if cohesin is destroyed in the absence of bipolar spindle forces. As a result, sister centromeres remain connected by DNA bridges, which are prone to breakage.

## Protection of centromeric Rec8 at metaphase II

A fundamental difference between the two deprotection models is the relationship between deprotection and the mechanisms that ultimately initiate anaphase II, namely the SAC in yeast and Emi2 in oocytes. We found that even in an unperturbed meiosis, yeast cells transiently activate the SAC at metaphase II. By inhibiting APC/$C^{Cdc20}$, the SAC inhibits deprotection of Rec8 as well as activation of separase as individual chromosomes biorient on the spindle. Upon silencing of the SAC, centromeric Rec8 is simultaneously deprotected on all chromosomes concomitantly with the activation of separase. Protection of Rec8 during metaphase II might be important, considering that Pds1/securin levels at metaphase II are lower than at metaphase I (Jonak *et al*, 2017). Indeed, meiosis II provides less time (40 min) for Pds1/securin accumulation than meiosis I (~ 3.5 h). While high Cdk1-cyclin B activity contributes to the inhibition of separase in vertebrates (Stemmann *et al*, 2001; Gorr *et al*, 2005), our data imply that Pds1/securin is the sole inhibitor of separase at metaphase II in yeast. The law of mass action therefore predicts that inhibition of separase is less robust at metaphase II. However, Pds1/securin accumulates to high levels when cells arrest at metaphase II due to inactivation of Cdc20. Even under these

conditions, experimental deprotection of Rec8 reduces the stability of centromeric cohesin. We propose, therefore, that at metaphase II, centromeric Rec8 needs to be protected against residual separase activity. In the absence of protection, even low levels of separase activity might threaten the integrity of dyad chromosomes because cleavage of centromeric cohesin is essentially irreversible. While expression of a non-cleavable Rec8 mutant during S phase blocks nuclear division at meiosis I and -II in a dominant manner (Katis *et al*, 2010), expression of the same Rec8 mutant during anaphase I has no detectable effect on nuclear division at meiosis II. It appears unlikely, therefore, that centromeric cohesin lost due to leaky separase activity can be replenished.

In oocytes, the SAC is silenced at entry into metaphase II, while APC/C$^{Cdc20}$ activity is inhibited, albeit not completely, by Emi2 until sperm entry (Suzuki *et al*, 2010). During the arrest, centromeric cohesin withstands bipolar spindle forces for long periods of time. Under these conditions, deprotection-by-tension would completely deprotect centromeric Rec8, and the integrity of dyad chromosomes would solely rely on the inhibition of separase. However, securin levels are much lower during the metaphase II arrest than at metaphase I (Marangos & Carroll, 2008; Nabti *et al*, 2008), probably due to the presence of significant APC/C$^{Cdc20}$ activity (McGuinness *et al*, 2009). Mitotic cells contain, in addition to securin, a separase inhibitor consisting of Sgol2 bound to Mad2. However, this inhibitor is unlikely to be active at the metaphase II arrest as its formation requires SAC signaling (Hellmuth *et al*, 2020). Furthermore, inhibition of separase by Cdk1-cyclin B has been observed at metaphase I but not at metaphase II (Nabti *et al*, 2008; Chiang *et al*, 2011). Thus, preserving the integrity of dyad chromosomes during the metaphase II arrest might require protection of centromeric Rec8 against leaky inhibition of separase. Consistent with this idea, Sgol2 and Mps1, required for protection at anaphase I, are also present during the metaphase II arrest (Lee *et al*, 2008; El Yakoubi *et al*, 2017).

The finding that yeast Sgol1-PP2A protects centromeric Rec8 when sister kinetochores are bioriented involves (at least) two puzzles: first, while sister kinetochore biorientation depends on cohesin complexes around centromeres, kinetochores are pulled all the way to the spindle poles. A recent analysis of chromatin structure around centromeres provides an elegant solution to this problem (Paldi *et al*, 2020). At least in mitotic cells, bioriented sister centromeres are held together by cohesin located on either side of, but not at, the centromere. As a result, centromeric DNA follows a chevron-like path with the kinetochore at the apex and cohesin sites at the ends. Second, while this configuration is consistent with the imaging of kinetochores and centromeric Rec8 at metaphase II, it raises the question of how or, indeed, whether Sgol1-PP2A associated with kinetochores protects centromeric cohesin. In mitotic cells, kinetochores have been shown to oscillate or "breathe" along the spindle axis (Goshima & Yanagida, 2000; He *et al*, 2000), which could create a cloud of PP2A activity that extends to cohesin. At metaphase II, however, sister kinetochore breathing is not apparent; kinetochores are located at the spindle poles for most of the time. A more likely hypothesis is, in our view, that there are different pools of Sgol1-PP2A and that Sgol1-PP2A colocalizing with kinetochores is not necessarily engaged in protecting Rec8. Indeed, Sgol has functions more directly linked to kinetochores, such as localizing the Aurora B/Ipl1 kinase and promoting SAC activity (Indjeian *et al*, 2005; Xu *et al*, 2009; Peplowska *et al*, 2014). On the other hand,

chromatin immunoprecipitations from metaphase I-arrested cells revealed a pool of Sgol1 that accumulates around centromeres in a pattern similar to that of Rec8 (Kiburz *et al*, 2005). While this fraction of Sgol1-PP2A is a better candidate for the Rec8 protector, it might be too small to be detectable by live imaging or chromatin spreading. In oocytes, Sgol2's functions at meiosis I also include kinetochore-related processes, such as the regulation of microtubule attachment through the inhibition of Aurora B/C kinases and the recruitment of the MCAK kinesin (Rattani *et al*, 2013). In addition, Sgol2 associates with centromeric and pericentromeric chromatin, which depends on the kinase activities of Mps1 and Bub1 (El Yakoubi *et al*, 2017). Since Mps1 is required for the protection of centromeric cohesin during anaphase I, it has been proposed that Mps1 regulates the Sgol2 pool involved in protection.

## Deprotection of centromeric Rec8

How is centromeric Rec8 deprotected, given that tension is not required? In yeast, Sgol1 and Mps1 are degraded by APC/C$^{Cdc20}$-dependent proteolysis at entry into anaphase II. Importantly, co-expression of nondegradable versions of both proteins delays cohesin cleavage until after spindle disassembly at anaphase II, leading to the formation of diploid spores (Arguello-Miranda *et al*, 2017; Jonak *et al*, 2017). Here, we have restored rapid removal of centromeric cohesin and nuclear division in cells arrested at metaphase II due to the lack of APC/C activity. This requires the deprotection of Rec8 through inhibition of Mps1 or removal of Sgol1 as well as the activation of separase by depletion of Pds1/securin. Together, these findings suggest that in yeast, APC/C$^{Cdc20}$ coordinates activation of separase with deprotection of centromeric Rec8 by mediating the degradation of Pds1/securin, Sgol1, and Mps1 at entry into anaphase II. This idea raises the question of how deprotection is confined to anaphase II. While protection at anaphase I and protection at metaphase II both depend on Sgol1, they differ in other aspects. In yeast, protection at anaphase I depends on Spo13 but not on Mps1 (Katis *et al*, 2004; Lee *et al*, 2004; Arguello-Miranda *et al*, 2017). At metaphase II, however, protection requires Mps1 activity, while Spo13 is absent. Spo13 and Mps1 might be part of the mechanism that switches protection from an APC/C-resistant state at meiosis I to an APC/C-sensitive state at meiosis II.

In oocytes, Sgol2 and Mps1 persist around centromeres during the metaphase II arrest (Lee *et al*, 2008; El Yakoubi *et al*, 2017), and we speculate that they maintain protection of centromeric Rec8. In contrast to yeast, Sgol2 and Mps1 persist on chromatin as sister centromeres separate (this work; Gryaznova *et al*, 2021). This suggests that Rec8 cleavage does not depend on complete removal of these proteins and raises the question of how protection might be disabled. We envision two scenarios whereby APC/C$^{Cdc20}$ elicits deprotection via inactivation of Cdk1-cyclin B. First, Cdk1 activity is required for Mps1's kinase activity and kinetochore localization (Morin *et al*, 2012; Hayward *et al*, 2019). Furthermore, in flies, Mps1 is inactivated through dephosphorylation by PP1 whose activity increases as that of Cdk1-cyclin B declines (Moura *et al*, 2017). Second, while Sgol1 protects centromeric cohesin from the prophase pathway in mitotic cells, its regulation might also apply to Sgol2. Sgol1 recruits PP2A not only to centromeric chromatin but also directly to cohesin, which depends on Sgol1's phosphorylation by Cdk1 (Liu *et al*, 2013a; Liu *et al*, 2013b). Thus, inactivation of Cdk1

deprotects centromeric cohesin by releasing only a small fraction of the Sgol1-PP2A detectable on chromatin. Interestingly, we noticed a change in the shape of the Sgol2 signal. At metaphase II, Sgol2 forms a crescent that extends from the kinetochore to the pericentromeric region where cohesin localizes. At anaphase II, however, Sgol2 is confined to the vicinity of kinetochores. Whether this change actually deprotects Rec8 remains to be investigated.

### Regulation of the SAC in meiosis

While the SAC is activated at each division in mammalian cells, it is rarely invoked during a normal cell cycle in yeast. We found, however, that yeast cells transiently activate the SAC at each division of an unperturbed meiosis. Thus, the small size of the spindle or a single microtubule per kinetochore in yeast does not necessarily imply that chromosome segregation is less reliant on the SAC. Surprisingly, cells unable to form MCCs, such as *mad2Δ* or *mad1,2,3Δ* mutants, still delay removal of centromeric cohesin in the absence of tension. Does this imply that tension, while not essential, plays an auxiliary role in deprotecting Rec8 or that tension regulates APC/C$^{Cdc20}$ activity even in the absence of MCC? Our data favor the latter possibility because lack of tension at meiosis II delays all APC/C$^{Cdc20}$-dependent processes in cells lacking MCCs. Work in worms, together with data from yeast and human cells, suggests that the Bub1–Bub3 complex recruits Cdc20 to the kinetochore protein KNL1 where it is exposed to two forms of tension-sensitive regulation: in the absence of tension, Cdc20 is incorporated into the MCC, whereas in the presence of tension, Cdc20 is dephosphorylated by PP1, which enhances its affinity for the APC/C (Yang *et al*, 2015; Kim *et al*, 2017; Bancroft *et al*, 2020). Thus, APC/C$^{Cdc20}$ activity is still responsive to tension in cells lacking MCCs. Remarkably, this MCC-independent regulation of APC/C$^{Cdc20}$ operates at meiosis II and probably also in mitosis (Proudfoot *et al*, 2019), but not at meiosis I. Whether sister kinetochore mono-orientation at meiosis I entails a stronger reliance of SAC activity on the MCC remains to be investigated.

# Material and Methods

### Construction of *Saccharomyces cerevisiae* Strains

We used diploid SK1 strains generated by mating of the appropriate haploids. Genotypes including the names of the fluorescent proteins are listed in Appendix Table S1. In the main text, all green and red fluorescent proteins are abbreviated as GFP or RFP, respectively. SK1 strains containing *REC8-mNeonGreen* (Arguello-Miranda *et al*, 2017), *CEN5-tetO* marked with TetR-tdTomato, or *P*$_{GAL1}$-*NDT80 P*$_{GPD}$-*GAL4-ER* for estradiol-inducible expression of Ndt80 (Matos *et al*, 2008) have been described. The non-cleavable phospho-mutant Rec8-24A has been characterized previously (Katis *et al*, 2010). PCR-generated cassettes were used for gene deletions (Goldstein & McCusker, 1999) and C-terminal tagging with fluorescent proteins (Sheff & Thorn, 2004), while integrative plasmids were based on YIplac vectors (Gietz & Sugino, 1988). To image microtubules, a plasmid expressing *mCherry-TUB1* from the *HIS3* promoter was integrated at *ura3*. To construct *cdc20*$^{ts}$-*mAR* strains, the SK1 *CDC20* gene (+1 to +1,928, <u>ATG</u> = +1) carrying the *cdc20-3* ts-mutation

(G360S; Shirayama *et al*, 1998) was cloned behind the *CUP1* promoter (450 bp). The *P*$_{CUP1}$-*cdc20-3* plasmid was integrated at *trp1* of cells in which endogenous *CDC20* is controlled by the mitosis-specific *CLB2* (1 kb) or *HSL1* promoter (500 bp). To deplete Ama1 at meiosis II, a plasmid expressing an *AMA1* cDNA from the early meiosis-specific *DMC1* promoter was integrated at *leu2* of *ama1Δ* strains (Arguello-Miranda *et al*, 2017). For auxin-inducible degradation, *PDS1* or *SGO1* were C-terminally tagged with AID* (Morawska & Ulrich, 2013) carrying a 30-residue linker. *PDS1-AID* and *SGO1-AID* diploids grow normally and produce spores with > 96% viability. Both alleles were crossed into strains harboring a *P*$_{CUP1}$-*OsTIR1-myc3* plasmid at the *ura3* locus (a gift from Neil Hunter; Tang *et al*, 2015). To introduce *mps1-as* into SK1, the ts-mutant SK1 *mps1-E517K* (Straight *et al*, 2000) was transformed with *mps1-M516G* DNA (Jones *et al*, 2005) followed by selection for growth at 37°C.

### Induction of meiosis in yeast

Meiosis was induced at 30°C as described (Matos *et al*, 2008). Briefly, colonies from YP-glycerol plates were evenly spread on YPD plates and grown to form a lawn (24 h). Cells were transferred to liquid YP-acetate medium (OD$_{600}$ ~ 0.3) and grown until reaching a transient G1-arrest (12 h, OD$_{600}$ ~ 1.6). Cells were washed with prewarmed sporulation medium (SPM, 2% K-acetate) and inoculated (OD$_{600}$ ~ 3) into 100 ml of SPM in a 2.8 l-Fernbach flask rotating at 200 rpm. *cdc20*$^{ts}$-*mAR* strains were induced to enter meiosis at 25°C. After arrest at metaphase I (8 h in SPM), cells were released into anaphase I with CuSO$_4$ (10 μM) for 50 min and then shifted to 37°C by placing the flask into a shaking water bath. 70 min after release ($t = 0$), cultures were treated with 1NM-PP1 (10 μM) and indole-3-acetic acid (IAA, 2 mM).

### Protein detection in whole cell extracts from yeast

Whole cell extracts prepared by glass bead extraction in 10% trichloroacetic acid were separated in SDS-8% polyacrylamide gels and transferred to PVDF membranes as described (Matos *et al*, 2008). Membranes were horizontally cut into 2–3 slices, which were incubated with primary antibodies for 2 h. Rabbit antibodies were used for detection of Ama1 (Oelschlaegel *et al*, 2005; 1:2,000), Ase1 (a gift from David Pellman; Juang *et al*, 1997; 1:1,000), Cdc20 (Camasses *et al*, 2003; 1:2,000), Clb3 (Schwickart *et al*, 2004; 1:3,000), Ndt80 (a gift from Kirsten Benjamin; Benjamin *et al*, 2003; 1:5,000), Pds1 (Katis *et al*, 2010; 1:500), Sgo1 (a gift from Adam Rudner; Lianga *et al*, 2013; 1:1,000), β-tubulin/Tub2 (a gift from Wolfgang Seufert; used in Oelschlaegel *et al*, 2005; 1:20,000), and Cdc5 (1:5,000), Dbf4 (1:5,000), Rec8 (1:10,000), and Spo13 (1:5,000) (Matos *et al*, 2008). We used goat antibodies from Santa Cruz for Clb1 (sc-7647; 1:300) and Clb4 (sc-6702; 1:400) and mouse monoclonal antibodies from Invitrogen for Myc (9E10; 1:1,000) and Pgk1 (22C5D8; 1:40,000). HRP-conjugated secondary antibodies were detected on X-ray films by ECL (GE Healthcare). Films were digitized using an Epson Perfection V750 Pro scanner.

### Microscopy of fixed cells and chromatin spreads from yeast

Immunostaining of formaldehyde-fixed cells was performed as described (Salah & Nasmyth, 2000). To prepare chromatin spreads

(Loidl *et al*, 1998), cells were collected and spheroplasted in buffers supplemented with Na-azide and 2-deoxyglucose (10 mM each). As primary antibodies, we used rabbit antibodies to GFP (1:200) and γ-tubulin/Tub4 (1:500) (Okaz *et al*, 2012) and monoclonal antibodies from mouse to γ-tubulin/Tub4 (Okaz *et al*, 2012; 1:100) and Myc (Invitrogen 9E10; 1:100) and from rat to α-tubulin (Serotec YOL1/34; 1:300) and Ha (Roche 3F10; 1:40). Alexa Fluor-conjugated secondary antibodies from donkey (Invitrogen) were used for detection. DNA was stained with DAPI. Cells were observed on a Zeiss Axio Imager. M2 with Colibri 7 LED light source and a Plan-Apochromat 100×/1.4 oil objective. Images were captured with a Zeiss Axiocam 506 mono camera controlled by ZEN 3.0 software and processed with Adobe Photoshop. The width of a single image is 10 μm. For quantifications, we scored 100 cells or ≥ 100 spreads per time point.

### Live imaging of meiosis in yeast

Meiotic cells were applied to an eight-well glass-bottom μ-slide (ibidi, Gräfelfing, Germany) coated with Concanavalin A in 250 μl of SPM to give a density of ~ 20 cells per field of view. Drugs and/or CuSO$_4$ were added in 50 μl of SPM. Cells were imaged on a Delta-Vision Ultra system composed of an Olympus IX71 microscope with auto focus, solid state illumination, UplanSApo 100×/1.4 oil objective, DeltaVision filters, CoolSnap HQ2 camera, and an environmental chamber set to 30°C. To film *cdc20$^{ts}$-mAR* cells, we used the Vaheat micro-heating system (Interherence, Erlangen, Germany). Cells were applied to the culture chamber in 150 μl of SPM at 25°C. CuSO$_4$ was added in 50 μl of SPM, and cells were shifted to 37°C over 4 min. In addition, the environmental chamber was set to 37°C 10 min before the shift. We acquired Z-stacks (8 × 1 μm) from 10 fields of view per strain in the GFP and the RFP channel every 10 min. Expose times were 100–300 ms with the adjustable ND filter set to 10% (GFP) or 32% (RFP). Z-stacks were deconvolved and projected (max intensity) to a 2D-image with softWoRx 7.0. Time-lapse series were produced in Fiji (http://fiji.sc/). The width of one frame is 5 μm. For quantification, meiotic events of individual cells were aligned to a reference event (e.g., Rec8 cleavage at anaphase I) set to t = 0 in each cell. Percentages of other events were calculated at 10 min intervals relative to the reference event.

### Oocyte harvest and culture

Mice were housed in animal facilities at Newcastle University, and all procedures were licensed (PPL 70/7960 and PPL PDD4CCF4F) by the UK Home Office under the Animal (Scientific Procedures) Act 1986. GV-stage oocytes were harvested from ovaries of 8–12-week-old CD-1 Swiss mice (Charles River) in M2 medium (Sigma) containing IBMX (200 μM) and then cultured in IBMX-free G-IVF PLUS medium (Vitrolife) for 16 h at 37°C in 6% CO$_2$, 5% O$_2$. Oocytes that produced a polar body were placed in activation medium (KSOM [Millipore] plus 5 mM SrCl$_2$ and 2 mM EGTA) for 1 h. Nocodazole (1 μM), reversine (1 μM), MG-132 (10 μM), and ICRF-193 (20 μM) were used at the indicated concentrations.

### Microinjection of oocytes with mRNAs and antibodies

For mRNA synthesis, cDNAs encoding Cdc20-wt, Cdc20-R132A, Securin-mNeonGreen, H2B-mScarlet, mNeonGreen-Sgol2, mScarlet-CenpC, and H2B-miRFP670 were cloned into pGEMHE (Liman *et al*, 1992). pGEMHE-eGFP-Trim21 and pGEMHE-eGFP-Trim21ΔC were gifts from Melina Schuh (Clift *et al*, 2017). Capped mRNA was synthesized using the mMessage mMachine T7 kit (Ambion), purified (RNeasy Micro Kit, Qiagen), and eluted in RNase-free water (~ 1 mg/ml). Metaphase II-arrested oocytes were injected with 5–8 pl of mRNA using pipettes preloaded with Fluorinert FC-770 (Sigma) and a pneumatic microinjector (Narishige IM-300). The plasma membrane was pierced with a single pulse (intensity ~ 5) from a piezo impact drive (Prime Tech PMM-150FU). Oocytes were injected in G-MOPS PLUS (Vitrolife) and then cultured in G-IVF PLUS medium. For Trim-Away of Rec8, GV-stage oocytes were injected with eGFP-Trim21 or eGFP-Trim21ΔC mRNA in M2 medium containing IBMX, allowed to recover for 2–3 h, and then cultured in G-IVF PLUS medium for 16 h. Metaphase II-arrested oocytes were injected with 5–8 pl of rabbit anti-mouse Rec8 antibody (a gift from Melina Schuh; Clift *et al*, 2017). The antibody (2 mg/ml) was buffer-exchanged into PBS by ultrafiltration (Amicon Ultra-0.5 100K, Merck) and diluted 1:1 with PBS containing 0.1% NP-40 and Alexa Fluor 647-labeled Dextran (4 mg/ml, Molecular Probes) as injection marker.

### Chromosome spreads from oocytes

To remove the zona pellucida, oocytes were briefly placed into acidified Tyrode's solution (pH 2.5, Merck) at 37°C. Oocytes were then placed in a hypotonic solution (0.5% Na-citrate) for 2 min and dropped along a Polysine adhesion slide (Thermo Scientific) onto a thin layer of fixative (1% paraformaldehyde/NaOH pH 9.2, 0.14% Triton X-100, 3 mM DTT, 100 mM sucrose) (Susiarjo *et al*, 2009). Slides were dried overnight at room temperature (RT) in a humidified chamber and stored at −20°C. For immunolabeling, slides were soaked in PBS for 15 min, washed twice for 2 min in 0.05% Photo-Flo (Kodak), and placed in blocking solution (PBS, 0.05% Triton X-100, 0.05% Tween 20, 10% normal goat serum) for 1 h at RT. Slides were then incubated overnight at 4°C with human anti-centromere antibodies (ACA, Antibodies Inc. 15-235, 1:50) and rabbit antibodies to topoisomerase II (Abcam ab109524, 1:100) or Sgol2 (a gift from José L. Barbero; Gomez *et al*, 2007; 1:100). After washing, slides were incubated for 1 h at RT with Alexa Fluor-conjugated anti-human (Molecular Probes A-21445, 1:400) and anti-rabbit (Molecular Probes A-11008, 1:800) secondary antibodies from goat. Slides were mounted with Vectashield plus DAPI (VectaMount) and imaged with a Plan-Apochromat 63×/1.4 oil objective on a Zeiss LSM 880 microscope with Airyscan. Z-stacks (35 × 0.16 μm) were acquired using Zen Black 2.3 software, and maximum-intensity projections were produced in Fiji.

### Live-imaging of oocytes

Oocytes were placed in 2-μl droplets overlaid with mineral oil (Sigma M5310) in a 4-chamber 35 mm glass-bottom dish (CellVis, Mountain View, CA) and imaged on a Zeiss LSM 880 microscope equipped with a C-Apochromat 40×/1.2 W objective, focus stabilization, and an environmental chamber set to 37°C in 6% CO$_2$, 5% O$_2$. To image securin-mNeonGreen, Z-stacks (30 × 2.5 μm) were recorded every 5 min. Mean projections were created for each time point and realigned with the Fiji StackReg plugin. Fluorescence

intensity over time was calculated with the Fiji Time-Series-Analyzer plugin from regions of interest (ROIs) drawn around each oocyte and an oocyte-free ROI for background correction. Imaging of mNeonGreen-Sgol2 was performed on a Zeiss LSM 880 with Airyscan, and chromosomes were tracked with MyPiC (Politi *et al*, 2018). Z-stacks (67 × 0.5 μm) were acquired at 5-min intervals shortly after GVBD.

## Statistical analysis

Mean values are given ± one s.d. Distributions were compared with two-tailed M–W tests and percentages with Fisher's exact test (two-tailed). Violin plots and Tukey-style boxplots were generated with BoxPlotR (Spitzer *et al*, 2014). The decline of Rec8-positive meiosis II cells follows an exponential decay ($R^2 \geq 0.94$) for data with $\geq 20\%$ Rec8-positive cells. To analyze rates of removal, linear regression was applied to log-transformed percentages of Rec8-positive cells over time, and slopes were compared with ANCOVA in GraphPad Prism 8.

# Data availability

This study includes no data deposited in external repositories.

**Expanded View** for this article is available online.

## Acknowledgements
We are grateful to Melina Schuh for generously providing reagents and advice to perform Rec8 Trim-Away. We thank José L. Barbero, Kirsten Benjamin, David Pellman, Adam Rudner, and Wolfgang Seufert for gifts of antibodies and Neil Hunter for yeast strains. We thank Orlando Arguello-Miranda for data in Fig EV4 and Isabella Mathes and Albena Bergsoy for help with tetrad dissection. This work was supported by the Max Planck Society (WZ) and by funding to MH from the European Union's (EU) Horizon 2020 research and innovation program under grant agreement No. 634113 (GermAge). Open Access funding enabled and organized by Projekt DEAL.

## Author contributions
VM, KJ, OL, JR, and IZ performed yeast experiments supervised by WZ. ML, LML, and CL performed oocyte experiments supervised by MH and WZ. WZ and MH designed research. VM, KJ, OL, JR, IZ, and WZ analysed yeast data. ML and WZ analysed oocyte data. WZ wrote the manuscript with input from all authors.

## Conflict of interest
The authors declare that they have no conflict of interest.

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
