## [Review Process File · The EMBO Journal]

Deprotection of Centromeric Cohesin at Meiosis II Requires APC/C Activity but not Kinetochore Tension

Wolfgang Zachariae, Valentina Mengoli, Katarzyna Jonak, Oleksii Lyzak, Mahdi Lamb, Lisa Lister, Chris Lodge, Julie Rojas, Ievgeniia Zagoriy, and Mary Herbert

DOI: [10.15252/embj.2020106812](https://doi.org/10.15252/embj.2020106812)

Corresponding author(s): Wolfgang Zachariae (zachar@biochem.mpg.de), Mary Herbert (mary.herbert@newcastle.ac.uk)

Review Timeline:

Submission Date:	16th Sep 20
Editorial Decision:	23rd Oct 20
Revision Received:	17th Dec 20
Editorial Decision:	19th Jan 21
Revision Received:	21st Jan 21
Accepted:	22nd Jan 21

Editor: Hartmut Vodermaier

Transaction Report:

Thank you for submitting your manuscript on centromeric cohesin deprotection in meiosis II to The EMBO Journal. We have now received reports from three expert referees, copied below for your information. As you will see, all referees consider the study and its findings important and technically well-done. We shall therefore be happy to consider this work further for EMBO Journal publication, following addressing of the various specific points raised by the reviewers through presentational changes and certain additional experiments where appropriate. In either case, our single-revision round policy will make it important to carefully respond to all comments at the time of resubmission. Should you have any questions related to answering the referees' points, or would require an extension of the default 90-day revision period, please do not hesitate to contact me. Finally, I would like to point out that our 'scooping protection' policy means that the publication of competing work during your revision would not affect our final decision on your study.

REFeree REPORTS

Referee #1:

Review on EMBO Journal manuscript 106812 entitled "Deprotection of Centromere Cohesin at Meiosis II Requires APC/C Activity but not Kinetochore Tension" by Mary Herbert, Wolfgang Zachariae and co-workers.

Shugoshin-PP2A protects (peri)centromeric cohesin from cleavage by separase, thereby preserving sister chromatid cohesion throughout meiosis I. As a corollary, separase-dependent sister chromatid separation in meiosis II requires prior inactivation of shugoshin-PP2A and, hence, deprotection of (peri)centromeric Rec8. The molecular mechanism of this deprotection is one of the most thrilling unresolved questions of meiosis research.

Previous work in yeast arrived at a "deprotection-by-degradation" model because it demonstrated shugoshin 1 (Sgo1) and the spindle assembly checkpoint (SAC) kinase Mps1 to be meiosis II-specific substrates of the ubiquitin ligase APC/C. In contrast, the temporal correlation of sister

kinetochore bi-orientation with the relocalization of shugoshin 2 (Sgo2) from centromeres to kinetochores in murine meiocytes gave rise to the "deprotection-by-tension" model, which essentially states that microtubule-dependent pulling forces contribute to shugoshin inactivation beyond SAC satisfaction and APC/C activation.

In the manuscript at hand, the Herbert- and Zachariae-labs test these two competing models by analyses in budding yeast and female murine meiosis. Their key findings are the following:

- While the absence of a bipolar spindle and, hence, the lack of tension across sister kinetochores profoundly inhibits anaphase II, the effects are due to activation of the SAC and can be fully attributed to the continued inhibition of the APC/C.
- When the separase inhibitor securin/Pds1 is degraded by auxin-induced degradation during a metaphase II arrest, centromeric Rec8 still persists. It disappears, however, when Sgo1 or Mps1 are inactivated (by auxin-induced degradation or chemical inhibition, respectively) concomitant to the destruction of securin. This not only demonstrates that separase becomes active under these conditions but also that yeast Rec8 continues to be protected by Sgo1 in presence of bipolar spindle forces.
- To study sister separation in the absence of tension during mammalian meiosis II, nocodazole-treated mouse oocytes (in which CSF was destroyed by Sr2+) were released into an anaphase-like state by SAC abrogation. This was achieved by expression of a Mad2-binding deficient Cdc20 variant or addition of the Mps1 inhibitor reversine. In both cases, about half of the dyads separated. Interestingly, Rec8-degradation by the Trim-Away-approach, which results in 90-98% separation in presence of spindle forces, is also less efficient (48%) in presence of nocodazole. This argues that Rec8 is fully cleaved upon APC/C activation and that spindle forces are merely required for decatenation and physical separation of sisters.

This is a very impressive study and a tremendous amount of work. Once more, the Zachariae lab establishes difficult and elegant synchronization methods and induced protein degradation procedures that are unprecedented in yeast meiosis research. The data are of the highest quality and all conclusions are fully supported by the existing data. Likewise, the mouse work is very convincing and falsifies a model, which has been dominating our view of mammalian shugoshin inactivation for 13 years. I have only few suggestions for further experiments and would not make acceptance of the manuscript dependent on their successful completion.

1) Interestingly, preventing MCC formation (e.g. by *mad2^Δ* still leaves anaphase II delayed in *spo12^Δ*, while restoration of bi-orientation (by additional *mam1^Δ*) restored wild type kinetics of sister chromatid separation (Fig. 3). The authors conclude that a MCC-independent (but SAC-dependent) mechanism contributes to the timely activation of APC/C in response to kinetochore biorientation and speculate about Cdc20 dephosphorylation. Given that Bub1 inhibits Cdc20 by (direct and indirect) phosphorylation (as demonstrated by the Yu lab), could it be tested whether additional inactivation of Bub1 would also advance anaphase onset in *mad2^Δ spo12^Δ* double mutants and render bi-orientation unnecessary?

2) An important aspect that is actually left unanswered by the study of Mengoli et al. is whether mammalian Rec8, like yeast Rec8, is also still protected from separase during a (prolonged CSF-mediated) metaphase II arrest of oocytes. An obvious experiment to address this would be the targeting of murine securin by Trim-Away. Unability of active separase to remove (peri)centromeric Rec8 and separate sisters in metaphase II would lend strong additional support to their claim that spindle forces do not contribute to inactivation of Sgo2. Therefore, the authors should try this experiment or - if they have already done so - explain why it did not work.

3) It would be nice to actually show the effect of decatenation. Could the authors repeat the

nocodazole > MG132 release in the absence of Rec8 (Fig. 8C) but this time add a topoisomerase II inhibitor, like ICRF193, along with the proteasome inhibitor?

Minor points:

p11/12: "Deletion of MAD2 in *spo12* cells also advances the removal of Rts1 foci; they now disappear at 46 +/- 12 min after anaphase I (Fig 3C and D). This suggests that once activated, APC/C does not require tension for its ability to remove Sgo1-PP2A from kinetochores at meiosis II. Importantly, *spo12 mad2* cells remove centromeric Rec8 ~80 min earlier than *spo12* cells, at 60 +/- 17 min after anaphase I (Fig 3B). In the absence of tension, centromeric Rec8 disappears in these cells without being separated from the bulk of PP2A/Rts1 during metaphase II (Fig 3C)." The last sentence reads like a contradiction to the rest of this paragraph!

p22: "In mitotic cells, cohesin is loaded onto centromeres even after S phase,..."
"In mitotic cells" should be replaced by "in mitotically dividing cells" to avoid confusion.

In some of the violine plots timing of events is hard to read. They would profit from better scaling.

CSF-arrested oocytes should exhibit high Cdk1 activity, which inhibits anaphase movements and, hence, would also be expected to counteract decatenation. Why then is Rec8 destruction sufficient to physically separate all dyads in a metaphase II arrest?

It would be nice to label (and maybe show an enlargement of) a "loose dyad" in one of the corresponding spreads.

Furthermore, it would be interesting to see an enlargement of the change in localization of Sgo2 from meta- to anaphase II (EV4B).

Referee #2:

In the manuscript by Mengoli and colleagues, the authors sought to determine the factors needed to release pericentromeric cohesin, testing several models in the field. By using a *spo12* mutant strain, they test whether tension is needed for the release of Rec8 protection. The *spo12* mutants do not duplicate their SPBs in meiosis II, making monopolar spindles, such that no attachments are under tension. In the mutant, removal of Rec8 is delayed, but this is due to a spindle checkpoint delay. By mutating *mam1* in *spo12*, chromosomes could biorient in meiosis II and cohesin was not protected. Deletion of the checkpoint protein *mad2* in *spo12* cells resulted in the cleavage of Rec8. These results suggest that there is a tension-independent but APC/C-dependent mechanism for Rec8 cleavage. However, the *spo12 mad2* cells did have a delay in APC/C activation in meiosis II. They find that activation of separase through the induced degradation of Pds1, together with the inhibition of Mps1, results in loss of Rec8 at metaphase II. These results suggest that both separase and Mps1 inhibition are needed for cohesin cleavage in budding yeast. They then perform experiments in mouse oocytes and show that centromeres can separate in nocodazole treated oocytes (without microtubules and without tension) when the SAC was over-riden, but not at all kinetochores, likely because of catenation. Inhibition of Mps1 did not further enhance the separation. These results suggest that the loss of protection of Rec8 due to APC/C activity is likely conserved.

Overall, this was a very thorough study, with high-quality experiments and analysis. There were

several important points learned from this study:

- 1) Their results do not support a tension-based model for deprotection of Rec8. Instead, their results support a model in which APC/C activation in meiosis II is needed for deprotection.
- 2) They developed a tool for meiosis II arrest and release
- 3) Cohesin is not reloaded or reinforced in meiosis II in budding yeast, the cohesin loaders are degraded in anaphase I.

I have only minor comments that can be addressed with writing changes:

- 1) The authors state that "This finding and additional considerations presented in Fig S2 argue that the role of sister kinetochore biorientation in deprotection be investigated in meiosis II". I am wondering what those additional considerations are? Interesting, Figure S2 shows that in the *mam1* mutant, only one spindle forms in meiosis II, but with the with deletion of *mad2*, two spindles form in meiosis II.
- 2) In the discussion, the authors state that, "This mechanism must be suppressed at metaphase II because it would result in premature tension-dependent deprotection of centromeric Rec8". The impact of this statement is unclear as stated. Could the authors refer back to their findings that separase is mostly but not completely inhibited by Pds1.
- 3) The Scc2-Scc4 figures are mislabeled on page 15. They should read Fig. 6A and 6B instead of 8A and 8B.
- 4) Is there any reason that Figure 1C is in greyscale and not overlaid?
- 5) Page 12, second paragraph, 1st sentence. They write that the SPO12 deletion causes a Mad2-dependent delay, but according to their data, the delay is partially (or mostly) Mad2 dependent.
- 6) On page 17, can the authors add a reference for the "loose dyads" that they are referring too?

Referee #3:

It is very well known for 20 years that at centromeres, cohesin's Rec8 subunit is protected from cleavage by separase at meiosis I and then deprotected to allow its cleavage at meiosis II. This stepwise protection mechanism is well conserved in eukaryotic species with sexual reproduction. However, little is known about the mechanism that governs deprotection during the second meiotic division. Two models have been proposed to explain the "open" susceptibility of Rec8 cohesin to be released at the anaphase II onset. The "tension model" propose that tension deprotect cohesin when chromosome congresses at metaphase II. The APC/C model proposes that APC/C coordinates deprotecton of Rec8 with separase activation. The work by Mengoli et al. address this issue by activating APC/Ccdc20 in the absence of sister kinetochore biorientation at meiosis II and shed some light into the mechanism of how deprotection of the centromeric cohesin at the second division take place in yeast and mouse oocytes. By the development of novel yeast meiotic mutants (that enable this sort of analysis), the authors conclude that protection of Rec8 by shugoshin and inhibition of separase by securin are both required for the stability of centromeric cohesin at metaphase II in yeast, which is quite accepted in yeast though not strictly demonstrated. The novelty relies on the fact that bipolar spindle forces are dispensable in both yeast and mouse for sister centromere separation. The yeast work is well conducted and the conclusions are supported by robust and novel experiments. The mouse work is some redundant in relation with the Sgo2 detailed description of its cytological localization and lacks the detailed genetic analysis of the yeast part (regardless of obvious experimental differences in both models). In summary, the MS is of interest and provides novel evidences to support that bipolar spindle forces are dispensable for the deprotection of centromeric cohesins at metaphase II in yeast.

Figure 1. By making use of *spo12Δ* that fail to reduplicate SPBs and thus contains a single SPB after anaphase I, the authors analyse PP2A^{Arts1} separation from centromeric Rec8 at metaphase II and show that it is dependent on spindle forces. Using this loss of tension model they also show that SAC is activated at the second division by indirectly visualizing MAD2-GFP. This activation is largely increased in the *spo12Δ* mutant as expected in a loss of tension model. Timing comparison between wild type and *spo12Δ* should be considered with caution given the intrinsic alteration of the SPB in this mutant and the pleiotropic function of *spo12*.

Figure 2. To demonstrate that SAC activity is the sole mechanism that delays entry into anaphase II in *spo12Δ* cells the authors try to restore sister kinetochore biorientation to eliminate the delay by mutating *mam1*. The results restore the kinetics similar to the wild type but with a partial delay. To make the comparison easier, it would be of help to compare the timing of the single and double mutant with the wild type. In summary, the model apparently seems not to be as clean as it is shown in the text and the efforts used to validate them are some cumbersome (*spo12Δ* *mam1Δ* double mutant).

Figure 3 is essential and especially Fig3C in which Rec8 disappears without being separated from the bulk of PP2A during metaphase II. These results are clearly validated by chromosome spreads (fig Ev2A) and could be moved to the main figure. Altogether these results are of great value and play against the guilty-by-proximity hypothesis.

Figure 4 shows the generation of a novel yeast strain that is able to arrest/release at metaphase II. Given the methodological-only relevance some panels could be moved to an Ev figure.

Figure 5. From this set of experiments the conclusion raised is clear and important in that both process of protection of Rec8 and inhibition of separase, safeguard centromeric cohesin when APC/C is inactive and sister kinetochores are bioriented at metaphase II. To validate this conclusion the Fig EV3D-F targeting Sgo1 for destruction is essential and could be moved to the general main figures at expenses of panel A.

Figure 6. The data generated in this figure is of biological relevance when extrapolated to human fertility. However, the position and the focus of the whole paragraph "The cohesin loader Scc2-Scc4 is destroyed at anaphase I" is out of focus in its actual context within the MS.

Figure EV4. The results described in the paragraph "APC/C activity induces sister centromere separation in mouse oocytes lacking microtubules" are known and have been published earlier using different approaches including video microscopy. The description is superfluous and completely lack novelty.

Figures 7-8. Experiments related to this figures are well conducted including the Trim21 destruction of Rec8 (already published by the Schuh's lab in Cliff et al., 2017). The conclusions raised support that bipolar spindle forces are not essential for the removal of centromeric Rec8 beyond silencing the SAC. Given the side effects of nocodazole, the use of an additional drug treatment to generate loss of tension would reinforce the conclusion.

Kinetochore individualization occurs onto sister kinetochores that have been previously mono-

oriented during prometaphase I. Though this process is not very well understood in vertebrates, Meikin together with Plk1 and partially through Sgo2 are known players involved in this process (Kim et al. 2015; PMID: 25533956). It would be interesting to analyze (as far as the Abs are available) how these proteins are modified when "deprotection" takes place under the different conditions.

Minor points

Figures figure 8A and 8B should be 6A and 6b in the paragraph "The cohesin loader Scc2-Scc4 is destroyed at anaphase I"

Finally, it is important to note that the only parallelism between the yeast and mouse work is that tension is not needed for deprotection (despite it is stated the high conservation mechanism between yeast and mouse) and this in part due to the lack of additional experiments to dissect which substrate of the SAC is/are involved in the deprotection of meiotic cohesins in mouse oocytes. Thus, the functional analysis of the deprotection mechanism is strongly biased to yeast meiosis.

Point-by-point response to reviewer's comments

We thank all reviewers for their insightful comments, which we feel have substantially improved the manuscript. Please find below our point-by-point response in blue text.

Referee #1:

Review on EMBO Journal manuscript 106812 entitled "Deprotection of Centromere Cohesin at Meiosis II Requires APC/C Activity but not Kinetochore Tension" by Mary Herbert, Wolfgang Zachariae and co-workers.

Shugoshin-PP2A protects (peri)centromeric cohesin from cleavage by separase, thereby preserving sister chromatid cohesion throughout meiosis I. As a corollary, separase-dependent sister chromatid separation in meiosis II requires prior inactivation of shugoshin-PP2A and, hence, deprotection of (peri)centromeric Rec8. The molecular mechanism of this deprotection is one of the most thrilling unresolved questions of meiosis research.

Previous work in yeast arrived at a "deprotection-by-degradation" model because it demonstrated shugoshin 1 (Sgo1) and the spindle assembly checkpoint (SAC) kinase Mps1 to be meiosis II-specific substrates of the ubiquitin ligase APC/C. In contrast, the temporal correlation of sister kinetochore bi-orientation with the relocalization of shugoshin 2 (Sgo2) from centromeres to kinetochores in murine meiocytes gave rise to the "deprotection-by-tension" model, which essentially states that microtubule-dependent pulling forces contribute to shugoshin inactivation beyond SAC satisfaction and APC/C activation.

In the manuscript at hand, the Herbert- and Zachariae-labs test these two competing models by analyses in budding yeast and female murine meiosis. Their key findings are the following:

- While the absence of a bipolar spindle and, hence, the lack of tension across sister kinetochores profoundly inhibits anaphase II, the effects are due to activation of the SAC and can be fully attributed to the continued inhibition of the APC/C.
- When the separase inhibitor securin/Pds1 is degraded by auxin-induced degradation during a metaphase II arrest, centromeric Rec8 still persists. It disappears, however, when Sgo1 or Mps1 are inactivated (by auxin-induced degradation or chemical inhibition, respectively) concomitant to the destruction of securin. This not only demonstrates that separase becomes active under these conditions but also that yeast Rec8 continues to be protected by Sgo1 in presence of bipolar spindle forces.
- To study sister separation in the absence of tension during mammalian meiosis II, nocodazole-treated mouse oocytes (in which CSF was destroyed by Sr2+) were released into an anaphase-like state by SAC abrogation. This was achieved by expression of a Mad2-binding deficient Cdc20 variant or addition of the Mps1 inhibitor reversine. In both cases, about half of the dyads separated. Interestingly, Rec8-degradation by the Trim-Away-approach, which results in 90-98% separation in presence of spindle forces, is also less efficient (48%) in presence of nocodazole. This argues that Rec8 is fully cleaved upon APC/C activation and that spindle forces are merely required for decatenation and physical separation of sisters.

This is a very impressive study and a tremendous amount of work. Once more, the Zachariae lab establishes difficult and elegant synchronization methods and induced protein degradation procedures that are unprecedented in yeast meiosis research. The data are of the highest quality and all conclusions are fully supported by the existing data. Likewise, the mouse work is very convincing and falsifies a model, which has been dominating our view of mammalian shugoshin inactivation for 13 years. I have only few suggestions for further experiments and would not make acceptance of the manuscript dependent on their successful completion.

We thank the reviewer for his/her positive comments on our study.

1) Interestingly, preventing MCC formation (e.g. by *mad2Δ*) still leaves anaphase II delayed in *spo12Δ*, while restoration of bi-orientation (by additional *mam1Δ*) restored wild type kinetics of sister chromatid separation (Fig. 3). The authors conclude that a MCC-independent (but SAC-dependent) mechanism contributes to the timely activation of APC/C in response to kinetochore biorientation and speculate about Cdc20 dephosphorylation. Given that Bub1 inhibits Cdc20 by (direct and indirect) phosphorylation (as demonstrated by the Yu lab), could it be tested whether

additional inactivation of Bub1 would also advance anaphase onset in *mad2Δ spo12Δ* double mutants and render bi-orientation unnecessary?

We agree with the reviewer that the tension-dependent but MCC-independent regulation of APC/C deserves further investigation. We would like to point out, however, that the molecular mechanisms of the SAC are not the focus of this work. We investigated the observation that tension promotes cohesin removal in the *mad2* mutant because it raises the possibility that tension, while not essential, has an auxiliary role in deprotection. In this case, the SPO12 deletion should delay cohesin removal but not spindle disassembly or cyclin degradation in the *mad2* mutant. We found, however, that the SPO12 deletion delays not only cohesin removal but also APC/C-dependent proteolysis in *mad2* mutants.

We agree that the Mad2-independent regulation of APC/C most likely involves Bub1. However, analyzing Bub1's function is complicated by the fact that it is also involved in recruiting Sgo1 to centromeres. Furthermore, Bub1 might either inhibit or fail to activate Cdc20 in the absence of tension, depending on the underlying mechanism: (1) Mammalian Bub1 phosphorylates and thereby inhibits Cdc20 in the absence of tension. According to this mechanism, inactivation of Bub1 in *mad2 spo12* cells should, indeed, eliminate the delay. However, it is unclear whether this mechanism exists in yeast in which Bub1's kinase activity is not required for SAC function. (2) Another mechanism is that Bub1-Bub3 bound to Knl1/Spc105 recruits Cdc20 to kinetochores and thereby exposes it to PP1/Glc7, which accumulates at Knl1/Spc105 in the presence of tension. Cdc20's dephosphorylation by PP1 increases its affinity for the APC/C. Since *mad2 spo12* cells lack tension, there is little PP1/Glc7 at Knl1/Spc105. Thus, preventing Cdc20's kinetochore recruitment by removal of Bub1 would have either little effect or even increase the delay in *mad2 spo12* cells. This pathway is evolutionarily conserved. It was discovered in *C. elegans* (Kim et al., 2017) and also exists in human and yeast cells (Bancroft et al., 2020; Yang et al., 2015). We briefly referred to this pathway in Results and explained it in more detail at the end of the Discussion. We have changed the wording in Results to make clearer that we speak about the PP1-dependent pathway and added the reference to the work in human cells.

Further analysis of this pathway in yeast is complicated by the fact that the kinase phosphorylating Cdc20 is currently unknown. While Cdc20 is phosphorylated by Cdk1 in mammals, this does not seem to be the case in budding yeast, at least at meiosis I and -II (our unpublished data). To circumvent this issue, we sought to activate Cdc20 by recruiting PP1/Glc7 directly to Cdc20. However, this approach was not successful in dephosphorylating Cdc20.

2) An important aspect that is actually left unanswered by the study of Mengoli et al. is whether mammalian Rec8, like yeast Rec8, is also still protected from separase during a (prolonged CSF-mediated) metaphase II arrest of oocytes. An obvious experiment to address this would be the targeting of murine securin by Trim-Away. Unability of active separase to remove (peri)centromeric Rec8 and separate sisters in metaphase II would lend strong additional support to their claim that spindle forces do not contribute to inactivation of Sgo2. Therefore, the authors should try this experiment or - if they have already done so - explain why it did not work.

We fully agree with the reviewer that activation of separase in metaphase II-arrested oocytes is an important experiment. Indeed, we tried Trim-Away of securin with the only commercially available antibody to mouse securin. While this monoclonal antibody recognizes securin in fixed oocytes, it was not effective in depleting a fluorophore-tagged securin expressed from micro-injected mRNA. Thus, performing this experiment has to await production of a new (polyclonal) antibody. Please note that antibodies to human securin do not cross-react with the mouse protein.

3) It would be nice to actually show the effect of decatenation. Could the authors repeat the nocodazole > MG132 release in the absence of Rec8 (Fig. 8C) but this time add a topoisomerase II inhibitor, like ICRF193, along with the proteasome inhibitor?

A failure to resolve DNA catenanes at anaphase II should result in ultrafine anaphase bridges (UFBs) between sister centromeres. To detect UFBs, we first looked at chromosome spreads from oocytes activated in the presence of ICRF193. Indeed, we can detect UFBs on some of these spreads. In particular, UFBs appear on spreads showing two well-separated groups of chromatids. However, these spreads are rare. While ICRF193 has little effect on sister centromere separation, it perturbs the orderly segregation of chromatids to the spindle poles (see **new Fig 9A**). As a result, most spreads show one field of scattered chromatids on which UFBs are hard to detect with certainty.

Nevertheless, we agree with the reviewer that evidence for topo II's involvement would be beneficial. Thus, we explored the notion that at least two mechanisms facilitate the topo II-dependent resolution of sister DNA catenanes. (1) since tension separates decatenated sister DNAs, it drives the catenation/decatenation reaction catalyzed by topo II towards decatenation. This mechanism is not available in the absence of microtubules. (2) By recognizing bent DNA, topo II is able to decatenate sister DNAs in an ATP-dependent process, called topological simplification. While

tension creates bent DNA at the catenation site, bent DNA also exists in the absence of tension, allowing topo II to decatenate sister centromeres at a low level. Indeed, we do observe fully separated sister centromeres upon APC/C activation in the absence of microtubules (Fig 7B, C). Importantly, inhibition of topo II should further reduce sister centromere separation induced by APC/C activity in the absence of microtubules. As shown in our new Fig 9, this is indeed the case: ICRF193 reduces sister centromere separation in oocytes treated with nocodazole, reversine, and SrCl₂ from 46 to 22% (Fig 9B). By contrast, ICRF193 has only a small effect on sister centromere separation in the presence of a spindle (Fig 9A). This is consistent with the notion that sister DNA catenanes cannot withstand bipolar spindle forces.

In addition, we have analyzed in more detail the behavior of sister sequences upon APC/C activation in the absence of tension in yeast. While mad2 spo12 cells readily remove centromeric cohesin (Fig 3A, B), chromatin spreads revealed that separation of CEN5-GFP sister dots is inefficient (51%). CEN5 sister centromeres stay close together at anaphase II (Fig EV2A), suggesting that their mobility might be restrained by DNA catenation. We have performed a new experiment with mad2 spo12 cells harboring GFP dots at sister sequences far away (394 kb) from CEN5 (new Fig EV2B). At anaphase II, the distance between these sister dots is, on average, four times larger than the distance between sister centromeres. Thus, the diffusion of sister centromeres is much more restricted than that of sister sequences on chromosome arms when centromeric cohesin is removed in the absence of bipolar spindle forces.

Minor points:

p11/12: "Deletion of MAD2 in spo12Δ cells also advances the removal of Rts1 foci; they now disappear at 46 +/- 12 min after anaphase I (Fig 3C and D). This suggests that once activated, APC/C does not require tension for its ability to remove Sgo1-PP2A from kinetochores at meiosis II. Importantly, spo12Δ mad2Δ cells remove centromeric Rec8 ~80 min earlier than spo12Δ cells, at 60 +/- 17 min after anaphase I (Fig 3B). In the absence of tension, centromeric Rec8 disappears in these cells without being separated from the bulk of PP2A-Rts1 during metaphase II (Fig 3C)." The last sentence reads like a contradiction to the rest of this paragraph!

In all our strains, removal of Rts1 foci is followed, 10-15 min later, by the cleavage of centromeric Rec8. These 10-15 min might be required for the phosphorylation of Rec8 and/or result from different detection limits of Rts1-GFP and Rec8-NeonGreen. Unfortunately, we are unable to image Rts1 foci and centromeric Rec8 in the same cells. The important difference is that in strains able to bi-orient sister kinetochores, centromeric Rec8 accumulates in the middle of the spindle at metaphase II, while Rts1 localizes at the kinetochore clusters at the poles. Thus, PP2A-Rts1 and centromeric Rec8 are spatially separated before entry into anaphase II. In strains unable to bi-orient sister kinetochores, centromeric Rec8 localized at the kinetochore clusters, in close proximity to PP2A-Rts1. Nevertheless, centromeric Rec8 is cleaved upon activation of APC/C-Cdc20. Thus, cleavage of centromeric Rec8 at entry into anaphase II requires APC/C activity but not spatial separation of centromeric cohesin and PP2A-Rts1 at metaphase II.

p22: "In mitotic cells, cohesin is loaded onto centromeres even after S phase,..."

"In mitotic cells" should be replaced by "in mitotically dividing cells" to avoid confusion.

We agree. This paragraph has been rewritten to make the rationale behind the Scc2 experiments clearer.

In some of the violine plots timing of events is hard to read. They would profit from better scaling.

All violin plots have been redrawn to show the same scale, and the "box" has been made thicker. The box plots (Figs S2 and S5) have twice the scale of the violin plots. All line graphs have a consistent scale.

CSF-arrested oocytes should exhibit high Cdk1 activity, which inhibits anaphase movements and, hence, would also be expected to counteract decatenation. Why then is Rec8 destruction sufficient to physically separate all dyads in a metaphase II arrest?

It has been shown in several systems that triggering anaphase in the presence of high Cdk1 activity does not inhibit the movement of chromatids per se but the coordinated and directed movement of sister chromatids to opposite spindle poles. If cohesion is destroyed while Cdk1 activity remains high, sister centromeres disjoin but chromatids then oscillate along the spindle axis and fail to form two distinct and well-separated sets of chromatids. This has been demonstrated, for instance, through live-imaging of artificial cohesin destruction by TEV protease in metaphase-arrested cells in the presence and absence of a Cdk inhibitor (Oliveira et al., 2010). Accordingly, Trim-Away of Rec8 in metaphase II-arrested oocytes triggers rapid sister centromere separation as shown first by Clift et al. (2017) and in

our **Fig 8A**, right). However, chromatids do not segregate into two distinct sets as would be the case during a normal anaphase II (**Fig 7A**, right). We would assume that the chromatid movement helps to resolve some but not all catenanes. However, ultrafine anaphase bridges cannot withstand spindle forces (Oliveira et al., 2010). It is unclear, therefore, whether the persistence of such bridges results in a discernible phenotype, given that chromatid segregation in the presence of high Cdk1 activity is already quite "messy".

It would be nice to label (and maybe show an enlargement of) a "loose dyad" in one of the corresponding spreads. We now show examples of normal and loose dyads in magnified cut-outs (**Fig 7C**, bottom).

Furthermore, it would be interesting to see an enlargement of the change in localization of Sgol2 from meta- to anaphase II (EV4B).

We have added magnified insets of the Sgol2 signals to **Fig EV5B** (formerly Fig EV4B).

Referee #2:

In the manuscript by Mengoli and colleagues, the authors sought to determine the factors needed to release pericentromeric cohesin, testing several models in the field. By using a *spo12* mutant strain, they test whether tension is needed for the release of Rec8 protection. The *spo12* mutants do not duplicate their SPBs in meiosis II, making monopolar spindles, such that no attachments are under tension. In the mutant, removal of Rec8 is delayed, but this is due to a spindle checkpoint delay. By mutating *mam1* in *spo12*, chromosomes could biorient in meiosis II and cohesin was not protected. Deletion of the checkpoint protein *mad2* in *spo12* cells resulted in the cleavage of Rec8. These results suggest that there is a tension-independent but APC/C-dependent mechanism for Rec8 cleavage. However, the *spo12 mad2* cells did have a delay in APC/C activation in meiosis II. They find that activation of separase through the induced degradation of Pds1, together with the inhibition of Mps1, results in loss of Rec8 at metaphase II. These results suggest that both separase and Mps1 inhibition are needed for cohesin cleavage in budding yeast. They then perform experiments in mouse oocytes and show that centromeres can separate in nocodazole treated oocytes (without microtubules and without tension) when the SAC was over-riden, but not at all kinetochores, likely because of catenation. Inhibition of Mps1 did not further enhance the separation. These results suggest that the loss of protection of Rec8 due to APC/C activity is likely conserved.

Overall, this was a very thorough study, with high-quality experiments and analysis. There were several important points learned from this study:

- 1) Their results do not support a tension-based model for deprotection of Rec8. Instead, their results support a model in which APC/C activation in meiosis II is needed for deprotection.
- 2) They developed a tool for meiosis II arrest and release.
- 3) Cohesin is not reloaded or reinforced in meiosis II in budding yeast, the cohesin loaders are degraded in anaphase I.

We thank the reviewer for his/her positive comments.

I have only minor comments that can be addressed with writing changes:

- 1) The authors state that "This finding and additional considerations presented in Fig S2 argue that the role of sister kinetochore biorientation in deprotection be investigated in meiosis II". I am wondering what those additional considerations are?

In the main text, we mentioned that at meiosis I, Mad2-GFP foci persist longer in *mam1* cells than in the wild-type (**Figs 1D and 2D**) and that metaphase I is prolonged in a Mad2-dependent manner in the *mam1* mutant (**Fig S2**). In the legend of **Fig S2**, we mentioned that *mam1* cells split CEN5-RFP sister dots less frequently at meiosis I than at meiosis II or at mitosis. In addition, we mentioned an elegant proposal from Nerusheva et al. (2014) that monopolin mutants contain bivalents with a bioriented and a mono-oriented pair of sister kinetochores, which results in spindle forces that cannot be balanced. We now present these findings and their implications in a single, shorter paragraph in the main text.

Interesting, Figure S2 shows that in the *mam1* mutant, only one spindle forms in meiosis II, but with the deletion of *mad2*, two spindles form in meiosis II.

Monopolin mutants fail to undergo nuclear division at meiosis I but reduplicate SPBs at metaphase II. With four SPBs in a single nucleus at meiosis II, monopolin mutants might be expected to assemble two spindle axes and to undergo a tetrapolar division that produces four equal-sized nuclei. However, in most of the cells, one spindle axis is more prominent, and segregates more of the chromosomes, than the other, which results in two larger and two smaller nuclei. While clearly detectable, centromeric Rec8 is not focused as sharply in these cells as in the wild-type or in *spo12 mam1* cells. As a result, *mam1* cells with one very prominent spindle axis are particularly attractive for making images showing strong centromeric Rec8 signals. We now provide images of more typical *mam1* cells in which the two spindle axes at meiosis II are discernible (**Figs 2B and S2**).

2) In the discussion, the authors state that, "This mechanism must be suppressed at metaphase II because it would result in premature tension-dependent deprotection of centromeric Rec8". The impact of this statement is unclear as stated. Could the authors refer back to their findings that separase is mostly but not completely inhibited by Pds1.

We have revised the relevant paragraph to improve clarity. It has been published previously that mitotic cells remove Sgo1 from centromeres/kinetochores at metaphase in a tension-dependent manner (Eshleman and Morgan, 2014; Nerusheva et al., 2014). Our data imply that Sgo1-PP2A/Rts1 is regulated in a different manner at metaphase II: at least a fraction of Sgo1-PP2A should persist on bioriented sister centromeres/kinetochores to promote the protection of centromeric Rec8.

To rigorously test this prediction, we have now performed live-imaging with temperature shift of *cdc20ts-mAR* cells containing RFP-labelled kinetochores and Rec8-GFP or Rts1-GFP (**new Fig 4C**). In these metaphase II-arrested cells, sister kinetochores are bioriented: centromeric Rec8 localizes between and Rts1 at the kinetochore clusters for long periods of time. This experiment shows that at meiosis II, tension resulting from sister kinetochore biorientation does not remove Sgo1-PP2A/Rts1 from kinetochores. Therefore, regulation of Sgo1-PP2A at metaphase II differs from that at metaphase of mitosis.

3) The *Scs2-Scs4* figures are mislabeled on page 15. They should read Fig. 6A and 6B instead of 8A and 8B.

We apologize for this mistake; it has been corrected.

4) Is there any reason that Figure 1C is in greyscale and not overlaid?

We have omitted the overlay in an attempt to save space. TetR-RFP bound to the *tetO* repeats at CEN5 creates a sharp dot, while free TetR-RFP generates a diffuse, nuclear signal. We use this diffuse signal to score nuclear division. In the overlay, however, this signal sometimes obscures the weak signals from centromeric Rec8. So, we feel that in these experiments, the overlay is dispensable, especially if space is limiting. This reasoning applies to all images showing CEN5-RFP sister dots together with Rec8-GFP (**Figs 1C, 2C, and 3A**). Since there is more space in the Appendix Figures, we show overlays and both channels in all these images (**Figs S2, S4, and S5**).

5) Page 12, second paragraph, 1st sentence. They write that the SPO12 deletion causes a Mad2-dependent delay, but according to their data, the delay is partially (or mostly) Mad2 dependent.

The Mad2-dependent delay (*spo12 vs. spo12 mad2*) is approximately 3-times longer than the Mad2-independent but tension-dependent delay (*spo12 mad2 vs. mad2*). To make this somewhat complicated issue more accessible, we introduce the two delays one after the other. First, we investigate whether deletion of MAD2 advances APC/C-dependent proteolysis but not cohesin removal (deprotection-by-tension) or whether it advances both processes (deprotection-by-APC/C). Then, we investigate whether the Mad2-independent but tension-dependent delay is or is not evidence of an auxiliary role of tension in deprotection.

6) On page 17, can the authors add a reference for the "loose dyads" that they are referring to?

We have coined this term to describe pairs of chromatids with adjacent but seemingly disconnected centromeres. We hypothesize that these dyads result from incomplete decatenation of centromeric sister DNAs. We now show examples of loose dyads in magnified cut-outs (**Fig 7C, bottom**). Furthermore, we now provide further support for this hypothesis in a new oocyte (**new Fig 9**) and a new yeast experiment (**new Fig EV2B**).

Referee #3:

It is very well known for 20 years that at centromeres, cohesin's Rec8 subunit is protected from cleavage by separase at meiosis I and then deprotected to allow its cleavage at meiosis II. This stepwise protection mechanism is well conserved in eukaryotic species with sexual reproduction. However, little is known about the mechanism that governs deprotection during the second meiotic division. Two models have been proposed to explain the "open" susceptibility of Rec8 cohesin to be released at the anaphase II onset. The "tension model" propose that tension deprotect cohesin when chromosome congresses at metaphase II. The APC/C model proposes that APC/C coordinates deprotection of Rec8 with separase activation. The work by Mengoli et al. address this issue by activating APC/Ccdc20 in the absence of sister kinetochore biorientation at meiosis II and shed some light into the mechanism of how deprotection of the centromeric cohesin at the second division take place in yeast and mouse oocytes. By the development of novel yeast meiotic mutants (that enable this sort of analysis), the authors conclude that protection of Rec8 by shugoshin and inhibition of separase by securin are both required for the stability of centromeric cohesin at metaphase II in yeast, which is quite accepted in yeast though not strictly demonstrated. The novelty relies on the fact that bipolar spindle forces are dispensable in both yeast and mouse for sister centromere separation. The yeast work is well conducted and the conclusions are supported by robust and novel experiments. The mouse work is some redundant in relation with the Sgo2 detailed description of its cytological localization and lacks the detailed genetic analysis of the yeast part (regardless of obvious experimental differences in both models). In summary, the MS is of interest and provides novel evidences to support that bipolar spindle forces are dispensable for the deprotection of centromeric cohesins at metaphase II in yeast.

We thank the reviewer for his/her comments.

Figure 1. By making use of *spo12Δ* that fail to reduplicate SPBs and thus contains a single SPB after anaphase I, the authors analyze PP2A^{ts1} separation from centromeric Rec8 at metaphase II and show that it is dependent on spindle forces. Using this loss of tension model they also show that SAC is activated at the second division by indirectly visualizing MAD2-GFP. This activation is largely increased in the *spo12Δ* mutant as expected in a loss of tension model. Timing comparison between wild type and *spo12Δ* should be considered with caution given the intrinsic alteration of the SPB in this mutant and the pleiotropic function of *spo12*.

We provide four lines of evidence (not just a single, indirect one) to support the notion that metaphase II is prolonged by the SAC in the *spo12* mutant: (1) persistence of Mad2-GFP at kinetochores is prolonged (**Fig 1D**). (2) Prolongation of metaphase II depends on the SAC component Mad2 (**Fig 3A, B**). (3) Prolongation of metaphase II is accompanied by a delay in the onset of APC/C-dependent proteolysis (**Figs S1 and S3**, formerly S1). (4) The delay is diminished by restoring sister kinetochore biorientation upon deletion of MAM1 (**Fig 2B-D**). It could be argued that by using four different approaches, we exercised great caution in reaching our conclusion that anaphase II entry in *spo12* cells is delayed due to a defect in spindle function, which in turn prevents sister kinetochore biorientation and consequently activates the SAC.

The term "pleiotropic" implies that Spo12 has functions beyond SPB reduplication that contribute to the phenotype we observed. Our results suggest that the deletion of MAM1 (a meiosis I-specific protein) restores the kinetics of meiosis II events to within 10 min of that of the wild-type. Furthermore, *spo12 mam1* double mutants produce (diploid) spores with a viability similar to that of (haploid) spores in the wild-type (91% vs. 94%, n = 144 spores per strain, p = 0.37, Fisher's exact test).

Figure 2. To demonstrate that SAC activity is the sole mechanism that delays entry into anaphase II in *spo12Δ* cells the authors try to restore sister kinetochore biorientation to eliminate the delay by mutating *mam1*. The results restore the kinetics similar to the wild type but with a partial delay. To make the comparison easier, it would be of help to compare the timing of the single and double mutant with the wild type. In summary, the model apparently seems not to be as clean as it is shown in the text and the efforts used to validate them are some cumbersome (*spo12Δ mam1Δ* double mutant).

As mentioned above, the MAM1 deletion largely restores the kinetics of meiosis II events in the *spo12* mutant to that in the wild-type. For instance, the median of the persistence of centromeric Rec8 is 55 min in *spo12 mam1* (**Fig 2B**) and 50 min in wild-type (**Fig 1B**). For spindle disassembly, the median is 100 min in both strains. The experiments shown in **Fig 1B-D** (wild-type vs. *spo12*) and **Fig 2B-D** (*mam1* vs. *mam1 spo12*) were performed as 4-strain experiments. Therefore, the data can be compared directly. Since our paper is directed not only at yeast meiosis experts, we sought to introduce our strategy step-by-step: **Fig 1** shows wild-type vs. *spo12*, and **Fig 2** shows

restoring sister kinetochore biorientation through deletion of MAM1. In the following figures (**Figs 3 and S4**), we actually use the three-strain comparison suggested by the reviewer.

Figure 3 is essential and especially Fig3C in which Rec8 disappears without being separated from the bulk of PP2A during metaphase II. These results are clearly validated by chromosome spreads (fig Ev2A) and could be moved to the main figure. Altogether these results are of great value and play against the guilty-by-proximity hypothesis.

We thank the reviewer for the comments on this set of experiments. Unfortunately, we find it difficult to squeeze the chromatin spreads into **Fig 3**, especially as this analysis has now been expanded to provide a comparison between the behavior of sister sequences at the centromere and 394 kb away from the centromere of chromosome 5 (**Fig EV2A and new Fig EV2B**).

Figure 4 shows the generation of a novel yeast strain that is able to arrest/release at metaphase II. Given the methodological-only relevance some panels could be moved to an Ev figure.

For the following reasons, we prefer to show these data in a main figure. (1) The metaphase II-arrest/release system is crucial to a key conclusion of our paper, namely that protection coexists with biorientation at metaphase II in yeast. (2) For the first time, this system allows to resolve meiosis I and meiosis II for biochemical approaches and is therefore of interest to others interested in comparing processes in meiosis I and -II. (3) While we would like to think that the metaphase II-arrest/release system constitutes a major advance, it is not perfect. It generates a culture containing "only" ~60% of metaphase II-arrested cells. This fact might be easily overlooked if the data were shown in an EV figure. Please note that in the following figures, we only analyze meiosis II-cells (showing 4 SPBs), which might give the impression of perfect synchrony.

Figure 5. From this set of experiments the conclusion raised is clear and important in that both process of protection of Rec8 and inhibition of separase, safeguard centromeric cohesin when APC/C is inactive and sister kinetochores are bioriented at metaphase II. To validate this conclusion the Fig EV3D-F targeting Sgo1 for destruction is essential and could be moved to the general main figures at expenses of panel A.

We have rearranged the panels of **Fig 5** to make the figure more accessible and to save space. We would prefer to keep the western blot (**Fig 5A**) in the figure. Showing that Pds1/securin has been successfully depleted, while cyclins remain stable, is crucial to the conclusions from this figure.

We thank the reviewer for emphasizing Sgo1's role in protection at metaphase II and hence its rigorous analysis. For this reason, we have now analyzed cells in which only Sgo1 is depleted at metaphase II. Similar to the inhibition of Mps1, depletion of Sgo1 decreases the half-life of centromeric Rec8 to 60% of the control value. This is consistent with the notion that Sgo1's ability to bind to centromeres/kinetochores and to protect Rec8 depends on Mps1. We now show the new SGO1-AID experiments together with the SGO1-AID PDS1-AID experiments in a new figure (**new Fig 6**).

Figure 6. The data generated in this figure is of biological relevance when extrapolated to human fertility However, the position and the focus of the whole paragraph "The cohesin loader Scc2-Scc4 is destroyed at anaphase I" is out of focus in its actual context within the MS.

We agree with the reviewer that we did a poor job in describing the rationale behind this experiment. We have revised the paragraph to make this issue clearer. We note that reviewer 2 found the Scc2 results noteworthy. Nevertheless, we have moved the data from a main figure to an EV figure (**new Fig EV4**, formerly Fig 6). The imaging of Scc2-GFP has been repeated (**Fig EV4A**), and the analysis of Scc2 protein levels has been supplemented with quantification of meiotic progression by immunofluorescence microscopy of fixed cells (**Fig EV4B, bottom**).

Figure EV4. The results described in the paragraph "APC/C activity induces sister centromere separation in mouse oocytes lacking microtubules" are known and have been published earlier using different approaches including video microscopy. The description is superfluous and completely lack novelty.

We respectfully disagree with this rather broad comment, which lacks references to the published work the reviewer deems relevant to our experiments.

Lee et al. (2008) have detected Sgol2 in metaphase II-arrested oocytes and claim that Sgol2 is pulled towards spindle poles and away from centromeric cohesin. The authors proposed that bipolar spindle forces deprotect Rec8 by spatially separating the protector Sgol2-PP2A from centromeric cohesin. We were intrigued by this idea (which we call the deprotection-by-tension model) because we and others did observe spatial separation of Sgo1 and PP2A-

Rts1/B56 from centromeric Rec8 in yeast. However, deprotection-by-tension has not been tested experimentally. Furthermore, spatial separation of Sgol2 and centromeric Rec8 at metaphase II was not apparent in a subsequent study on mouse oocytes (Chambon et al., 2013). We consider it important, therefore, to investigate Sgol2's localization at meiosis II and to show the relevant data. In our hands, neither live-imaging (**Fig EV5A**, formerly EV4A) nor chromosome spreading (**Fig EV5B**, formerly EV4B) of mouse oocytes revealed relocation of Sgol2 from the pericentromere (where centromeric cohesin localizes) towards spindle poles at metaphase II. We have now added to **Fig EV5B** a representative spread from metaphase II-oocytes treated with nocodazole. It shows a clear reduction in inter-kinetochore distances, thereby confirming that our spreading procedure preserves the kinetochore configuration of dyad chromosomes under tension.

The paragraph mentioned by the reviewer also includes **Fig 7**. We do not claim to be the first to show spreads from metaphase II-arrested or SrCl₂-activated oocytes. However, these spreads are important controls characterizing our experimental system and its quantification. These experiments are subsequently used to overturn a long-held model of the regulation of sister centromere separation at meiosis II in mouse oocytes. In this situation, we think it essential to demonstrate, with images and quantification, that our experiments are properly controlled.

Figures 7-8. Experiments related to this figures are well conducted including the Trim21 destruction of Rec8 (already published by the Schuh's lab in Clift et al., 2017). The conclusions raised support that bipolar spindle forces are not essential for the removal of centromeric Rec8 beyond silencing the SAC. Given the side effects of nocodazole, the use of an additional drug treatment to generate loss of tension would reinforce the conclusion.

We are happy to learn that the reviewer appreciates our Rec8 Trim-Away experiments. We have cited the Clift et al. paper from Melina Schuh's group in Results and in Material and Methods. Indeed, we are very grateful to Melina Schuh for advice and crucial reagents to perform these experiments (see Acknowledgements). For the following reasons, we are reluctant to use animals to repeat our experiments with another microtubule inhibitor. (1) We used nocodazole at a concentration (1 micro-molar) typical for experiments involving microtubule depolymerization in mouse oocytes and are not aware of side effects relevant to our conclusions. (2) In all experiments involving nocodazole, the control and the experimental group have been treated with nocodazole (**Fig 7B, C; Fig 8B, C; Fig 9B**). (3) From **Fig 8C**, it can be concluded that transfer of nocodazole-treated oocytes to nocodazole-free medium restores efficient sister centromere separation.

Kinetochore individualization occurs onto sister kinetochores that have been previously mono-oriented during prometa-metaphase I. Though this process is not very well understood in vertebrates, Meikin together with Plk1 and partially through Sgo2 are known players involved in this process (Kim et al. 2015; PMID: 25533956). It would be interesting to analyze (as far as the Abs are available) how these proteins are modified when "deprotection" takes place under the different conditions.

This comment is unclear to us. Our work does not investigate kinetochore individualization or orientation. We analyze APC/C-dependent proteolysis and sister centromere disjunction in yeast cells and mouse oocytes arrested at meiosis II. Meikin and its yeast homologue Spo13 are meiosis I-specific proteins, which are destroyed in an APC/C-dependent manner at anaphase I and do not reappear at meiosis II (Kim et al., 2015; Sullivan and Morgan, 2007). Indeed, we have detected Spo13 to show that APC/C-mediated proteolysis at meiosis I is normal in spo12 mutants (**Figs S1 and S3**, formerly S1).

Minor points:

Figures 8A and 8B should be 6A and 6b in the paragraph "The cohesin loader Scc2-Scc4 is destroyed at anaphase I".

We apologize for this mistake, which has been corrected.

Finally, it is important to note that the only parallelism between the yeast and mouse work is that tension is not needed for deprotection (despite it is stated the high conservation mechanism between yeast and mouse) and this in part due to the lack of additional experiments to dissect which substrate of the SAC is/are involved in the deprotection of meiotic cohesins in mouse oocytes. Thus, the functional analysis of the deprotection mechanism is strongly biased to yeast meiosis.

We have stated that the general principles of protection appear conserved, a notion that is widely accepted and described in essentially all reviews on the topic. As elaborated in our Introduction and Discussion, it is remarkable in

this regard that two fundamentally different models have been proposed for reversing protection, that is, for deprotection. We have investigated deprotection-by-APC/C and deprotection-by-tension in yeast, and our data are consistent with the former but not the latter. Since deprotection-by-tension has originally been proposed for mouse oocytes, but has not been tested experimentally, we investigated the predictions of this model in oocytes. Our data are inconsistent with deprotection-by-tension. We agree with the reviewer that we did not reveal the actual mechanism of deprotection in oocytes. Nevertheless, our work suggests that yeast might serve as a valuable hypothesis-generator to eventually decipher the mechanism in mammalian oocytes.

AUTHOR AMENDMENT OF RESPONSE LETTER / 6/1/2021

An alternative to the experiment on the cohesin loader Scc2-Scc4 in our manuscript EMBOJ-2020-106812, which was valued very differently by the reviewers:

The background is the following: we had shown that in yeast, two mechanisms safeguard centromeric cohesin at metaphase of meiosis II, inhibition of separase by securin and protection of Rec8 from separase by Shugoshin-PP2A. We have argued that these two mechanisms are important because at metaphase II, very small amounts of cohesin withstand bipolar spindle forces, while loss of cohesin due to leaky separase inhibition might be irreversible. However, this argument is only valid if centromeric cohesin cannot be replenished. We had shown that the cohesin loader Scc2-Scc4 is destroyed at anaphase I, making it unlikely that centromeric cohesin can be reinforced at meiosis II. This experiment was appreciated by reviewer 1 and in particular by reviewer 2. However, reviewer 3 felt that it disrupts the flow of the manuscript. In the revision, we have better explained the reasons for addressing what we consider an important issue and moved the loader experiment to an EV figure (Fig EV4A and B).

Nevertheless, I would like to bring to your attention a more direct experiment to test whether centromeric cohesin can be reinforced, namely by expressing at anaphase I a non-cleavable version of Rec8. The result is clear: non-cleavable Rec8-24A does not perturb meiosis II when expressed at anaphase I. By contrast, we have shown previously that the Rec8-24A mutant blocks both divisions in a dominant manner when expressed during S phase. The advantages of this experiment are that it does not depend on any assumption regarding the mechanism (e.g. dependence on Scc2-Scc4) and that it uses a well-characterized reagent, namely the non-cleavable Rec8-24A mutant, which we have analyzed in detail in two previous papers. Furthermore, this experiment does not introduce any new molecules (like Scc2-Scc4).

I have attached a PDF with the alternative figure, the legend, and the relevant text for the Results section. I am grateful for your time and consideration.

Alternative Figure EV4. Expression of non-cleavable Rec8 at anaphase I.

CDC20-mAR cells released from the metaphase I-arrest ($t = 0$) were treated with estradiol ($t = 50$ min, arrows) to induce expression of wild-type Rec8 or the non-cleavable phospho-mutant Rec8-24A from an estradiol-dependent promoter at anaphase I. Top, immunoblot analysis of whole cell extracts. Separate activity at meiosis II generates cleavage products from wild-type Rec8 but not from Rec8-24A. Bottom, immunofluorescence microscopy of fixed cells. Non-cleavable Rec8-24A has no effect on nuclear division at meiosis II. Data are representative of two independent experiments.

The paragraph below would replace the paragraph entitled “The cohesin loader Scc2-Scc4 is destroyed at anaphase I” in the main text.

Yeast cells cannot replenish centromeric cohesin at meiosis II

Are protection of Rec8 and inhibition of separase the sole mechanisms safeguarding centromeric cohesion at metaphase II? A possibility worth considering is that meiotic cells replenish centromeric cohesion at meiosis II. While cohesion is established during DNA replication in an unperturbed cell cycle, it is reinforced at G2/M in response to DNA damage (Strom et al., 2007, Unal et al., 2007). To test whether centromeric cohesin can be reinforced at meiosis II, we used an estradiol-inducible promoter to express the Rec8-24A mutant at anaphase I (Fig EV4). When expressed at entry into meiosis, this mutant is capable of generating normal sister chromatid cohesion but is resistant to cleavage by separase due to the lack of phosphorylation sites for the Rec8-kinases Hrr25 and Cdc7-Dbf4 (Arguello-Miranda et al., 2017, Katis et al., 2010). Thus, expression of Rec8-24A during S phase blocks nuclear division at meiosis I and -II in a dominant manner (Katis et al., 2010). By contrast, Fig EV4 shows that expression of Rec8-24A at anaphase I has no detectable effect on nuclear division at meiosis II. It appears unlikely, therefore, that centromeric cohesin can be replenished after anaphase I in yeast.

Thank you again for submitting your revised manuscript to The EMBO Journal. Both referees 2 and 3 have now looked at it once more, and I am pleased to say found all concerns well-addressed and the study suitable for publication. Before proceeding with formal acceptance and production, there are at this point only a few editorial issues to be taken care of, as listed below.

REFEREE REPORTS

Referee #2:

Overall, this revision has addressed all of the concerns in my original review with clarification of

several points, and the addition of experiments and figures. I feel that the paper is very thorough and is experimentally very rigorous. The findings are an important contribution to the field.

Regarding the alternative Figure EV4:

The authors prepared an alternative figure to the Fig. EV4. The goal was to ask whether cohesins are replenished during meiosis II. Originally, they performed experiments with the cohesin loader Scc2-Scc4, showing that it is no longer present after meiosis I. In this alternative figure, they express a non-cleavable Rec8 mutant in anaphase I and show that nuclear division still occurs in meiosis II, supporting their previous conclusion that cohesin was not re-loaded in meiosis II. I find that this figure is more straight-forward than the Scc2-Scc4 findings. I would favor replacing the original figure with the alternative figure.

Referee #3:

The authors have answered and addressed all the points raised by the three reviewers and conducted new experiments and modified the MS accordingly. The resulting revised version is now ready to be directly accepted for publication in EMBO J.

Response to reviewer comments.

1. In the legend and text accompanying Fig 4, we have changed the designation of the metaphase II-arrest/release strains from cdc20ts-mAR and CDC20-mAR to cdc20ts-mAR ama1 and CDC20-mAR ama1, respectively. This reflects the fact that these strains contain an Ama1-depletion allele. The new nomenclature is consistent with our previous work (Arguello-Miranda et al., 2017) and with Fig EV4, in which CDC20-mAR strains (containing wild-type AMA1) have been used.

Thank you for submitting your final revised manuscript for our consideration. I am pleased to inform you that we have now accepted it for publication in The EMBO Journal.

Corresponding Author Name: Mary Herbert & Wolfgang Zachariae

Manuscript Number: EMBOJ-2020-106812